

# Understory vegetation relationships with soil element contents in a northern boreal forest ecosystem near a phosphate massif

Laura Matkala[1], Maija Salemaa[2], Jaana Bäck[1]

[1]Institute for Atmospheric and Earth System Research / Forest Sciences, Faculty of Agriculture and Forestry, PO Box 27, 00014, University of Helsinki, Helsinki, Finland

[2] Natural Resources Institute Finland (Luke), Latokartanonkaari 9, 00790 Helsinki, Finland

*Correspondence to*: Laura Matkala (laura.matkala@helsinki.fi)

**Abstract.** We studied the relationship of forest understory vegetation with nutrient contents of soil and tree leaves near Sokli phosphate ore in northern Finland, where the soil contains naturally high variation in phosphorus (P) contents. At most study plots boreal dwarf shrubs, bryophytes and lichen formed a dense mat under a mixture of sparsely growing *Pinus sylvestris*, *Picea abies* and *Betula pubescens*. However, some plots were dominated by *B. pubescens* and had a higher variety and number of forbs and grasses in the understory. The total P content in the soil humus layer explained the abundance and species composition of the vegetation slightly better than the total nitrogen content. The spatial variation in contents of soil elements was high both between and within plots, emphasizing the heterogeneity of soil. High contents of P in the humus layer (max. 2600 mg kg$^{-1}$) were measured from the birch-dominated plots. As the P contents of birch leaves and leaf litter were also rather high (2580 mg kg$^{-1}$ and 1280 mg kg$^{-1}$, respectively), this may imply that the leaf litter of birch forms an important source of P to the soil.

## 1 Introduction

Climate and availability of soil nutrients are important factors controlling the composition of tree stands and understory vegetation in boreal forests (Cajander 1909, 1949, Kuusipalo 1985, Økland and Eilertsen 1996). High latitude forest ecosystems are characteristically nutrient-poor and modified by low temperatures and short growing season. Cold climate affects decomposition rate of detritus and the amount of nutrients released from organic material (Hobbie et al. 2002). The availability of nutrients in soil, and utilization of nutrients by plants play a critical role in many functions of forest ecosystems (Merilä and Derome 2008). The edaphic conditions are reflected in the growth and chemical composition of plant species, as well as in species composition of vegetation (Vinton and Burke 1995, Salemaa et al. 2008). In addition, tree species affect the understory vegetation by shading (Verheven et al. 2012, Tonteri et al. 2016) and regulating nutrient input in throughfall precipitation (Salemaa et al. 2019) and leaf litterfall (Ukonmaanaho et al. 2008).

Nitrogen (N) and phosphorus (P) are generally the growth-limiting nutrients for plants (Koerselman and Meuleman 1996). Boreal forests are mostly N-limited (Tamm 1991), and fertilizing with N usually speeds up forest growth (Saarsalmi and





Mälkönen 2001). Nitrogen is bound in organic material, and only little is directly available for plants as inorganic ammonium ($NH^{4+}$) and nitrate ($NO^{3-}$) (Marschner 1995) or as organic forms like amino acids (Näsholm et al. 2008 and references within). The primary source of N is the atmosphere, while P originates from the weathering of bedrock (Walker and Syers 1976, Vitousek et al. 2010). Phosphorus is tightly bound in the soil (Marschner 1995, Hinsinger 2001), and plants take up P directly

only as ortophosphates, which are compounds formed from inorganic and organic P in processes requiring specific phosphatase enzymes (Jackman and Black 1952, Deiss et al. 2018). Phosphorus deficiency occurs in temperate and tropical forest ecosystems, but P is rarely a limiting factor in boreal upland forests (Augusto et al. 2017). However, P can be growth-limiting on boreal peatlands (Moilanen et al. 2010, Brække and Salih 2002). It has been demonstrated, that the interaction of soil N and P is significant for forest growth on a global scale (Augusto et al. 2017). Hedwall et al. (2017) found that the species richness

of vascular plants doubled with combined NP fertilization in southern Sweden, but not when either of the nutrients was added alone. This positive effect was strongest in grass species. In boreal, N-limited forests the number of vascular plant species (grasses and forbs) increased with increasing N concentration of the organic layer (Salemaa et al. 2008). Hofmeister et al. (2009) noticed that in a temperate forest the herb layer diversity was higher at P rich than P poor soils, but only if strong N limitation occurred simultaneously at the P rich soils. However, in many regions, where humans have enhanced atmospheric

N deposition, the number of plant species has decreased (Dirnböck et al. 2016). For instance, high soil N was related to decreased herb layer species richness in deciduous forests in Sweden (Dupré et al. 2002).

In this study, we analysed whether plant species composition and tree leaves indicate soil nutrients in a research site in northern boreal zone (Hämet-Ahti 1981) in Sokli, Finland. At this site, the soil contains naturally large variations in P contents. In Sokli,

there is a large deposit of phosphate rock, a carbonatite complex mainly consisting of apatite [$Ca_5(PO_4)_3F$], which was discovered by the Mining and Steel Company of Rautaruukki Oy in 1967 (Vartiainen and Paarma 1979). Plans to open a phosphate mine in Sokli have been on display for decades and will possibly be realized in the future. The vegetation at the carbonatite complex differs from that of the typical forests of the region (Talvitie 1979, Pöyry Environment 2009). Downy birch (*Betula pubescens*) is dominating and often the single tree species, whereas the more typical forests of the region are

dominated by Scots pine (*Pinus sylvestris*) or Norway spruce (*Picea abies*). Understory vegetation at Sokli is somewhat rich in herb and grass species compared to the typical forests, where dwarf shrubs, bryophytes and lichen dominate the understory. Similar vegetation as in Sokli grows as patches elsewhere in the region. An additional factor affecting vegetation composition at our research site is reindeer herding. Bryophytes have replaced many lichen species (Väre et al. 1995, Susiluoto et al. 2008, Akujärvi et al. 2014, Köster et al. 2018), and the number of seedlings of broadleaved trees has been found to decrease (Kreutz

et al. 2015) in forests where reindeer trampling and grazing occurs. All our plots were located in areas where reindeer roam freely.

The general aim of this study was to investigate the undisturbed state of the forest ecosystem in the Sokli area, for the possible situation there is a need to monitor the effects of phosphate mining. Vegetation, soil and foliage chemistry surveys provide



data on the current state of the ecosystem (from the year 2015) that can be used as a reference level for the changes. Phosphate mining can cause, for instance, aerial deposition of heavy metals and phosphate to the surroundings of the mine (Reta et al. 2018). The combined effects of mining activities, reindeer grazing and climate change can lead to unpredictable changes on the element cycles. Particularly, we investigated the relationship between the understory vegetation and soil chemical composition in the study area and how the element compositions of tree leaves are related with soil chemistry.

Because N and P are known to be important limiting elements for biomass production of plants (Vitousek et al. 2010) we hypothesize, that

a)  N and P contents of the soil humus layer correlate with the abundance and species composition of the understory vegetation

b)  N and P contents of needle and leaf biomass reflect total N and P contents of the soil humus layer

## 2 Material and methods

### 2.1 Site description

We established 16 study plots along four transects around the planned Sokli mining district (67° 48' N, 29° 16' E) in Savukoski, eastern Lapland in 2014 and 2015 (Fig. 1). No plots were located inside the mining district. The carbonatite massif of Sokli belongs to the Devonian Kola Alkaline Province (KAP) (Tuovinen et al. 2015). Nine of the plots were located in Natura 2000 conservation areas. Plots A4, A5 and A6 were on Värriö, A1 and A2 on Yli-Nuortti, B1, B2 and B3 on Törmäoja and D5 on UK-puisto – Sompio – Kemihaara Natura area. The Törmäoja and Yli-Nuortti Natura areas have carbonatite in the soil, which explains the occurrence of grass species in understorey vegetation and the sparse, birch-dominated tree cover. By topography, Törmäoja is a valley, reminding the form of a kettle (kattilalaakso in Finnish) (NATURA 2000 - Standard Data Forms FI1301512 and FI1301513). The mid parts of the valley are treeless, or the trees are at sapling stage, because cold winds blowing through the valley kill the new buds in the spring. Our plots at Törmäoja were on the edge of the less steep western part, where some mature trees grew. Plots A3 and D2 were also birch-dominated with grass species in the understory, but they were not on any Natura area. Majority of the plots had a mixed composition of at least two tree species, but some were dominated by only one species. The plots of this study together with the SMEAR 1 station (Station for Measuring Ecosystem-Atmosphere Relations) at Värriö Subarctic Research Station (67° 46' N, 29° 35' E) (Hari et al. 1994) serve as a gradient type network for monitoring the current status and the possible, mining-induced, changes of the environment in the future.

Meteorological parameters from the years of data collection and for climatological normal period of 1980–2010 are presented in Table 1. Wind blows almost equally from southwest and northeast during spring and summer, whereas in winter and autumn the prevailing wind direction is southwest (Ruuskanen et al. 2003). Growing season, when daily average temperature exceeds 5 °C lasts from June to September. Soils are haplic podzols with sand tills (FAO 1988).





## 2.2 Plot and sub-plot set-up and field work

The distance between two plots depended on topography and existing roads, but generally, it was about two kilometres. A plot consisted of four clusters, each including three square-shaped sub-plots sized 1 m² for observations and sampling (Fig. 2). The size of the whole plot was 30 x 30 m.

### 2.2.1 Sampling of vegetation and soil

We recorded the species of all trees growing on the plots and measured their heights and diameters (at height 1.3 m). Stem volumes were estimated using the equations of Laasasenaho (1982). We assessed visually the cover (%) and counted the number of plant species in the understory vegetation in all 12 sub-plots per plot in summers 2014 and 2015. We used a 1 m² square frame to delineate the sub-plot (Salemaa et al. 1999). All species in the bottom layer (bryophytes and lichens) and field layer (dwarf shrubs, tree seedlings, grasses, sedges and forbs) were included. Altogether sixteen soil samples were collected from each sixteen plots using a soil corer in June 2015. The soil samples were taken close to the vegetation, the maximum distance being approximately one meter (cf. Liski 1995). The samples were separated by visual criteria into four soil horizons; the top layer, which is a mixture of litter and decomposing organic layer (F), the humus layer (O), the elluvial layer (A), and the illuvial layer (B) (cf. Köster et al. 2014). The actual humus layer was very shallow and the soil rocky, which made the sampling difficult in some plots. In some plots it was, thus, possible to sample only the upper soil layers. The soil samples from each horizon were combined into composite samples in each cluster already in the field. The composite samples were air-dried except for organic F and O horizons, which were dried at 60 °C for 48 hours. Dried mineral soils were sieved with a 2 mm sieve and samples from F and O horizons milled before storing them in a dry place for further analyses.

### 2.2.2 Sampling of needles and leaves

We collected foliar samples from all plots in 2015. We sampled both pine and spruce needles, as well as green birch leaves and birch leaf litter lying on the ground. Five pines and five spruces per plot were chosen for needle sampling in September. If less than five trees per species were present, all of them were chosen. Three branches (length approximately 50 cm) were taken from upper third of the canopy, using a branch saw. We took only second order branches because cutting of first order branches would have been too destructive to trees (cf. Helmisaari 1990). Needle age classes (C = current year, C+1 = one-year-old, C+2 = two-years-old needles) were separated from each branch and dried at 65 °C for 48 hours, milled and stored in a dry place for further analyses. The samples were combined so that there was one C, one C+1 and one C+2 composite needle sample per tree.

We sampled green birch leaves in July and leaf litter in September 2015. Approximately 10 green leaves from 10 different trees were picked and combined, totalling of 100 leaves per plot (Rautio et al. 2010). Only mature, undamaged leaves were chosen. Birch litter was collected under the same trees where the green leaf samples were taken and approximately the same



number as green leaf samples. We aimed to take litter leaves shed in the current year, so that they were decomposed as little as possible. Green and litter leaves were dried in a similar way than needles and manually cleaned from extra material, such as soil particles and needles. After that, they were milled and stored in a dry place for further analyses.

## 2.3 Laboratory analyses

Total element contents of potassium (K) and P were analysed from all soil and foliar samples by inductively coupled plasma emission spectrometry (ICP-OES). For this analysis, the samples were first wet combusted. One gram of mineral soil samples and 0.3 g of organic samples were combusted with 1 ml of $H_2O_2$ and 10 ml $HNO_3$ and heated in a microwave oven. The samples were then filtered with Whatman Grade 589/3 filter paper and stored in plastic bottles in a cooler until analysed.

Total carbon (C) and N were analysed directly from dried and milled foliar samples as well as from F and O soil layers. Two to three mg of sample was measured and analysed with VarioMax analyser. Soil pH was measured from two O layer samples per plot and their mean value used. 20 mg of dried sample was mixed together with MilliQ water (50 ml). The suspension was covered and left standing for 24 hours, and pH was measured with a glass electrode.

## 2.4 Statistical analyses

We used one-way ANOVA and Tukey's post-hoc test for analysing the plot-wise differences in the soil and needle element contents. In the latter case, plot-averages of needle elemental contents were calculated across all needle age classes and both conifer species. One-way ANOVA was also used in analysing differences between the needle age classes.

We calculated plot-wise averages of the plant species coverages in the sub-plots. We ordinated this vegetation data by global
non-metric multidimensional scaling (NMDS) (Minchin 1987) using the Vegan package (Oksanen et al. 2018) in R programme 3.4.3 (R Development Core Team, 2017). Ordination pattern of the study plots and weighted averages of plant species were analysed to find the main environmental gradients behind the vegetation variation. We analysed the data in three-dimensional space, but present the results in 1 vs. 2 and 1 vs. 3 dimensions. We then fit the plot-averages of soil elements (contents from O horizon), some other environmental variables as well as species numbers as linear vectors to the ordination pattern of sample
plots. The correlation between the environmental variables and the ordination was calculated by a linear vector procedure (envfit in Vegan). The soil total P in the O horizon was also fitted as smooth surface on the ordination pattern in order to analyse the form of the relationship (linear or non-linear). The fit was done by generalized additive model (GAM, Gaussian distribution error).



## 3 Results

### 3.1 Soil element contents

The average contents of total P in  different soil horizons are presented in Fig. 3. and the average N and C:N in two organic soil layers (F and O horizons) are presented in Fig. 4. The outlying points in Fig 3. as well as the high standard deviations of

5  P imply rather high variation between and within plots (Fig. 3 and Fig. 4, Table A1). Other soil elements showed similar variation. The significant differences between plots are presented in Tables A2, A3 and A4. In general, topsoil had the highest P content, but at many plots, also deeper soil layers had high P content. Certain plots (A1, A3, B1, D2) had distinct P and N content, and C:N ratios than most other plots. The N contents and C:N ratios were in most cases higher in F than O horizon. Soil N:P ratios were similar across the plots. Majority of the plots had higher N:P ratio in the F horizon than O horizon.

### 3.2 Needle and leaf element contents

The average contents of elements are presented in Table 2. Needle P contents were highest in the C needles, and significantly different from other age classes in both pine and spruce (Table B1). Unlike the expectations, the needle P contents of both species were rather similar across plots (Table B2). On the other hand, N and C contents, as well as C:N  ratio of the conifers showed some between-plot variation (p < 0.05). Foliar N:P ratio did not show any differences in either species between plots.

Spruce had slightly higher needle P contents than pine in all age classes, whereas N contents seemed to go vice versa between the species. Birch had highest P contents of green leaves compared to other species. Also leaf litter of birch had quite high P contents and in general more variation in element contents than the green leaves. Nitrogen contents were lower in leaf litter than in green leaves, but the contents of C increased slightly from green leaves to litter. Green leaves had significantly higher contents of elements than leaf litter, but no differences between the plots were discovered. Figure 5 presents a correlation table

including soil elements in the O horizon together with leaf element contents and number of species in the understory. Birch K (green leaves) correlated with soil K and pH, litter N with soil N:P and litter K with soil N, but otherwise no correlations between foliar element contents and soil element contents were found. Number of species in the understory correlated with soil elements but not with foliar elements.

### 3.3 Ordination analysis of understory vegetation

Figure 6a-d depicts how the plots were related to each other and how the weighted averages of the plant species were located in the ordination space (dimensions 1 vs. 2 and 1 vs. 3). The closer the plots were to each other, the more similar their vegetation was. Plots located more on the left-hand side had higher number of forbs and grasses growing on them than the plots on the right-hand side (Fig.6a). Species such as *Calamagrostis epigejos*, *Carex spp.*, *Rubus arcticus* and *Luzula pilosa* had relatively high coverage on the plots on the left. The plots further on the right had more species, which tolerate poor and dry growing

conditions, such as *Cladonia* and *Cladina* lichens. The tree species (Table 3) also changed from right to left, as the plots on the right were dominated by pine, whereas furthest on the left in plot D2 birch was the only tree species present. In general,



the fertility trend in the vegetation followed the first dimension, while the moisture gradient followed the second dimension. Moisture demanding species, such as *Equisetum sylvaticum* and *Rhododendron tomentosum* were located in the upper part of Fig. 6b., and those tolerating drier conditions, such as *Peltigera rufescens* and *Stereocaulon tomentosum*, were located in the lower part of the ordination space in Fig. 6b. Another moisture gradient, expressing specific paludified conditions seemed to follow the third dimension. Peatland species like *Sphagnum angustifolium* and *Aulacomnium palustre* were located on the upper part, and species preferring dry conditions, such as *Cetraria islandica*, were in the lower part of Fig 6d. Considering all three dimensions of ordination space, the generalist species, such as *Polytrichum commune*, *Pleurozium shcreberi* and *Vaccinium myrtillus* were located in the middle.

The vector arrows fitted to the ordination space depict the correlations between environmental variables and sample plot ordination (Fig. 7c-d). The length of an arrow indicates the magnitude and the direction the polarity (plus–minus) of correlation. The correlation values between the ordination pattern and different explanatory vectors are given in Table 4. The highest correlations occurred between the P content of soil O horizon and the ordination pattern. The isocline gradient of soil P in relation to the ordination pattern was almost linear (Fig. 7a). Vectors of soil pH, N and P content all increase towards the more fertile plots, but the vectors of soil C:N and N:P went to the opposite directions (Fig. 7b) indicating poor soil conditions. The average total number of grass, forb and sedge species in the study plots also increased towards the more fertile plots (Fig. 7 d).

## 4 Discussion

The spatial variation in soil element contents between clusters was very high at some plots, emphasizing the heterogeneity of soil fertility level. The P contents of soil samples in our study (180–2600 mg kg$^{-1}$ in the O horizon) fell mostly in the category we could expect based on literature. Mäkipää (1999) reported values between 800–2100 mg kg$^{-1}$ for P content in humus layer of forest soil in southern Finland. The P contents from the top 5 cm of soil in northern Finland varied widely in both Naruska (385–1970 mg kg$^{-1}$) and Pallas (599–3030 mg kg$^{-1}$) in a study by Reimann et al. (1997). Mikkola and Sepponen (1986) found high variation in P content from organic soil in Kilpisjärvi, northwestern Finland, with highest values at around 800 mg kg$^{-1}$. Most of our plots had highest P content at organic soil layers, implying that decaying plant parts were a major source of P. Low arctic soils tend to have organic P as the primary form of P (Weintraub 2011). The content of organic P usually gets smaller in the deeper soil (Achat et al. 2009). Thus, if P content is high in deep soil layers, as it was in some of our plots, the source of P is most likely in the underlying bedrock. The plot-wise average pH of our soil samples agreed to that measured by Köster et al. (2014). The pH of soil humus layer correlated positively with the number of grass, herb and sedge species, which is reasonable, since higher pH usually implies a more fertile site. The soil N contents from our plots agreed with the reported values from Finnish forest sites (Merilä and Derome 2008, Salemaa et al. 2008) ranging between 9.8–12.8 g kg$^{-1}$. Salemaa et al. (2008) reported soil C:N ratio of 40 from northern Finnish forest site, which is higher than what we measured. Total element contents in soil do not tell how much of the nutrients are in plant-available form. They can, however, be used for estimating



the long-term nutrient status of forest soil (Mäkipää 1999), and as proxies of available pools at a regional scale, especially on older soils (Liptzin et al. 2013, Augusto et al. 2017).

The P and N levels of our needle samples were similar than previously measured in Finland (Helmisaari 1990, Merilä and Derome 2008, Moilanen et al. 2013). The mean P contents of our pine C needles were within the deficiency range of 1200–1500 mg kg$^{-1}$, which Brække (1994) reported for Norway spruce and Scots pine. The mean spruce needle P contents were within the pre-optimum range of 1500–1800 mg kg$^{-1}$. Both pine and spruce needles had N contents falling in the deficiency range. The N contents of both green birch leaves and leaf litter agreed with those reported by Ferm and Markkola (1985). The P contents of green leaves were higher than the approximately 2000 mg kg$^{-1}$ they measured, which is also considered as the

deficiency limit (Miller 1983). Our litter P contents were near the approximately 1500 mg kg$^{-1}$ that Ferm and Markkola (1985) measured from a 40-year-old forest, but much less than those reported from younger forests. Birch leaves were a major source of litter at the plots where soil P was high. Viro (1955) found that the leaf litter of birch had remarkably high P content compared to other Finnish tree species. In a litter experiment in Abisko (northern Sweden), the addition of birch litter increased both the total P (Sorensen and Michelsen 2011) and the available P (Rinnan et al. 2008) contents in the organic soil layer at

those subarctic heaths, where *H. splendens* dominated the moss layer.

All the plant species, which grew on our plots, were common Finnish forest species (e.g., Reinikainen et al. 2010, Finnish Biodiversity Info Facility 2018), and most plots resembled each other in their plant species composition. We found evidence that the richness of understory vegetation was more related to soil P content than to soil N content. Both soil P and N contents

correlated with the abundance and species composition of understory vegetation, which supports our first hypothesis. We also found soil C:N ratio correlating negatively with the abundance and species composition in the understory. Soil C:N ratio was an important variable explaining aboveground species richness also in a deciduous forest in north-western Germany (Schuster & Diekmann 2005). Salemaa et al. (2008) studied connections between understory vegetation and the nutrient concentrations of soil organic layer at several sites in Finland. They found soil N concentration and C:N ratio the most important nutrient

variables explaining site vegetation patterns. They also measured extractable soil P, which showed highest concentrations on the plots located in northern Finland, and seemed to have more power in explaining vegetation patterns in northern Finland compared to southern Finland. Soil P availability was one of the key factors in plant community variation in alpine habitats in Troms, northern Norway (Arnesen et al. 2007).

Our second hypothesis stated that the N and P contents of foliar biomass reflect N and P contents of soil, but our results did not support this hypothesis. The reason could be that we measured total contents of N and P in soil instead of plant available contents of these nutrients. What should also be noted is that needles were sampled at different time of year than soil, and both needle (e.g. Helmisaari 1990) and soil nutrient contents vary along the seasons. For instance, snowmelt can cause release of P in the spring (Weintraub 2011). Our soil samples were taken a couple of weeks after the snowmelt. On the other hand in the

early summer soil contained less litter than in the autumn. However, as all soil sampling was conducted at the same time of the season and all needle sampling at the same time of the season, the comparison between the plots is not hindered.

As soil microbial activity may change due to warmer climate, N may become more available from organic sources at high latitudes in the future (Rustad et al. 2001). This, together with high soil P may induce growth and affect vegetation dynamics. Climate change has already caused variation in the vegetation at high latitudes, as deciduous shrub coverage has expanded at the Arctic region (Sturm et al. 2001, Park et al. 2016). Greater deciduous shrub cover causes increased leaf litter input, which in turn may bring more nutrients that are recyclable to the ecosystem.

## 5 Conclusions

We found that the total P content of the soil humus layer is an important factor explaining understory vegetation dynamics at our research plots near the Sokli phosphate ore in Finnish Lapland. The plots with high soil total P in the humus layer (max. 2600 mg kg$^{-1}$) had birch as the dominating tree species. Downy birch leaf litter has been discovered to contain large contents of P, so it is possible that the leaf litter from birch caused the high total P contents in the humus layer. The mean P content of our birch litter samples was 1280 mg kg$^{-1}$, which is higher than the P contents of C+1 and C+2 needles of pine (1150 and 1160 mg kg$^{-1}$, respectively). Most of the plots with high total P in the humus layer had high total P contents also in the B layer, where the maximum content was as high as 5500 mg kg$^{-1}$. It is interesting that in our study the soil total P explained the understory vegetation dynamics better than soil total N did, as usually N is considered more important for understory vegetation in boreal forests. As climate change and the possible mining activities may affect the nutrient and vegetation dynamics in the studied region, the research that we carried out has an important part in both clarifying the current situation and forming a baseline for evaluating the magnitude of changes in the future.

## Data availability

We have made all data used in the analyses publicly available. All can be downloaded at: https://doi.org/10.23728/b2share.615b46018cef40fe8c9d9245c56f0547.

## Author contributions

LM and JB planned the study set-up; LM conducted all fieldwork, laboratory analyses and statistical analyses and led the writing process; MS had a substantial role in guiding the ordination analyses and the writing process; all authors contributed to writing.



## Competing interests

The authors declare that they have no conflict of interest.

## Acknowledgements

We thank the staff at the Värriö Subarctic Research Station for providing a full board during the field work, Marjut Wallner
for guidance at the laboratory, Jarkko Isotalo for commenting on the statistical analyses, Jukka Pumpanen and Kajar Köster
for advice and equipment for the soil sampling and Olli Peltola for helping with field work. The study was supported by Maj
and Tor Nessling Foundation and Finnish Center of Excellence grants 272041 and 307331.

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

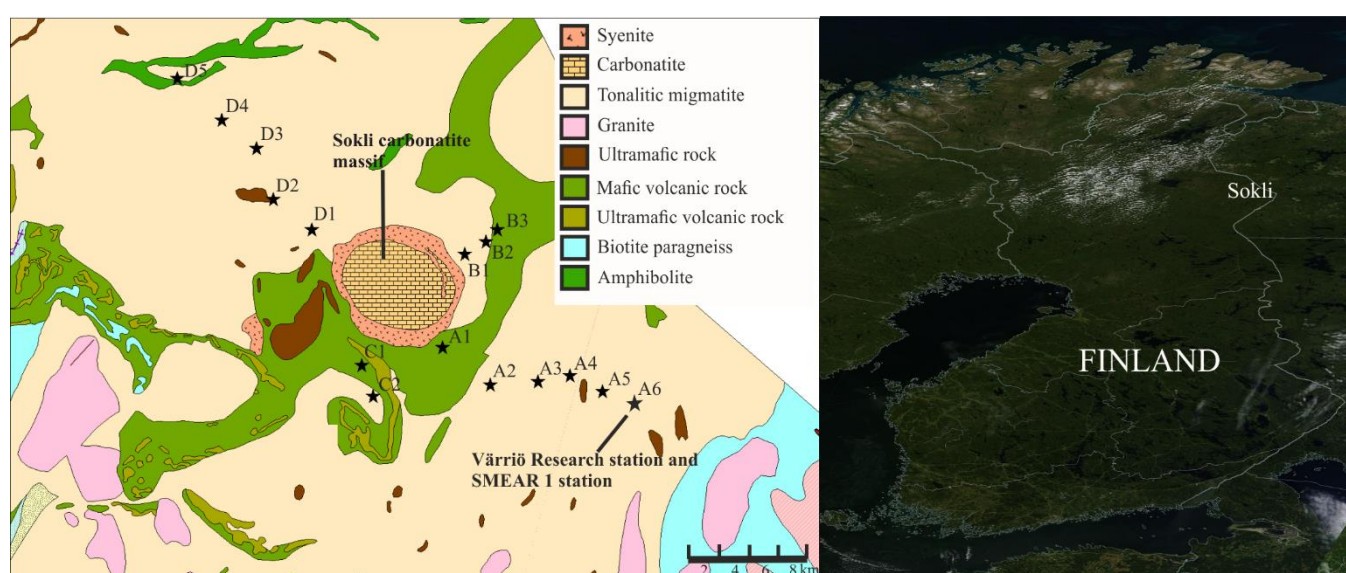

**Figure 1: On the left a geological map of the research area, where plots are marked with black stars. The easternmost plot is located at the SMEAR 1 station (Geologigal map from Hakku Service, satellite image from NASA).**





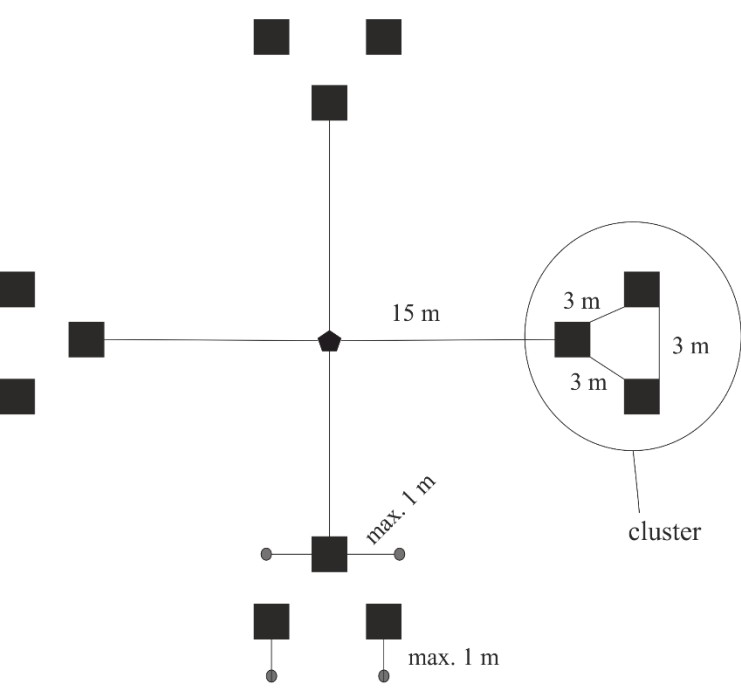

**Figure 2: The set-up of each research plots with clusters and sub-plots within clusters.**





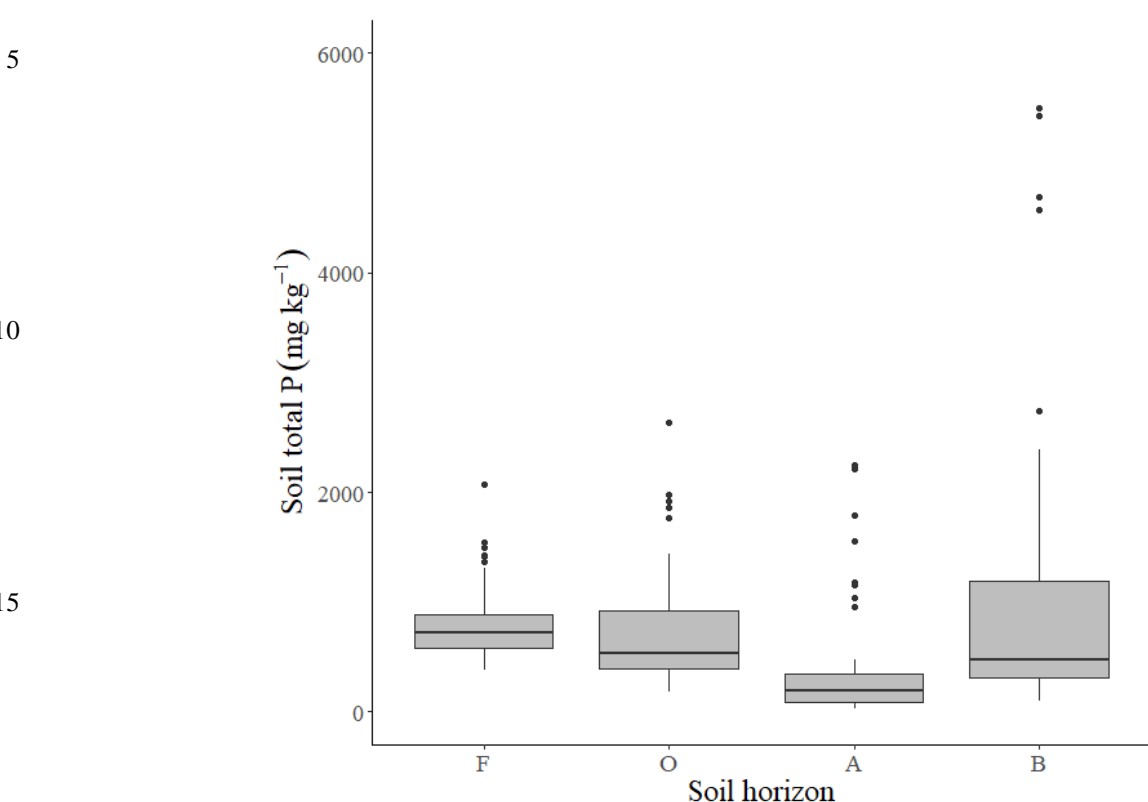

**Figure 3. Soil total P contents in different soil horizons. The lines inside boxes denote medians, lower and upper hinges are the first and third quantiles, the whiskers cover values ranging 1.5 x the inter-quartile range (IQR, the distance between the first and third quartiles) from the hinge and the points outside are outliers not fitting inside the previously mentioned range.**



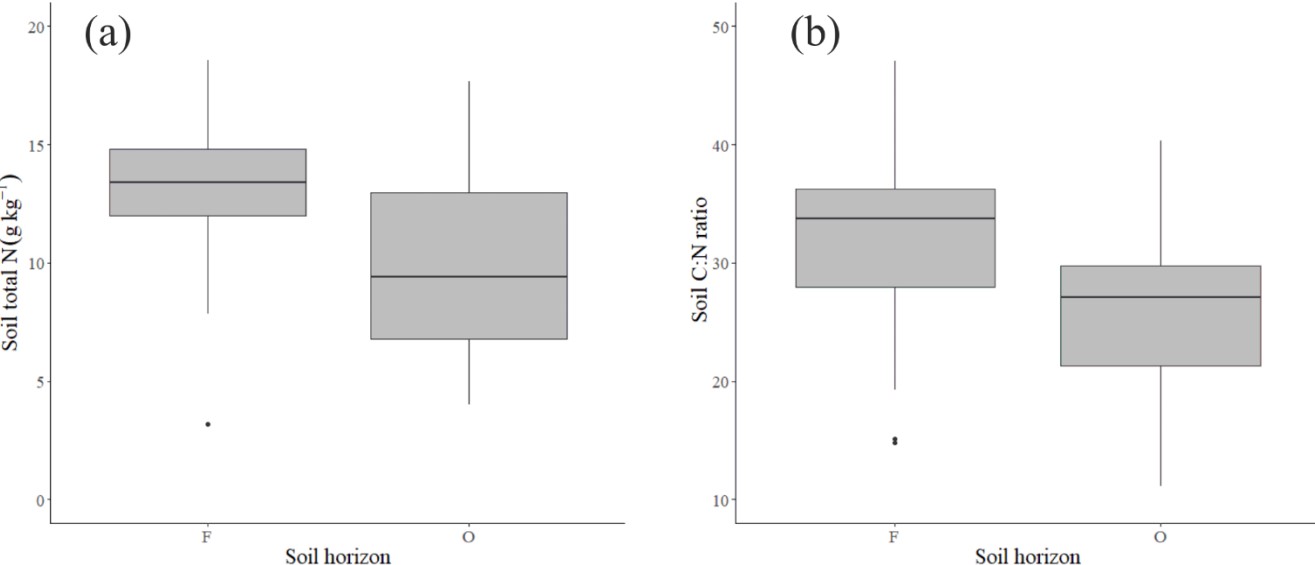

**Figure 4. Soil total N content (a) and soil C:N ratio (b) in soil horizons F and O. The lines inside boxes denote medians, lower and upper hinges are the first and third quantiles, the whiskers cover values ranging 1.5 x the inter-quartile range (IQR, the distance between the first and third quartiles) from the hinge and the points outside are outliers not fitting inside the previously mentioned range.**





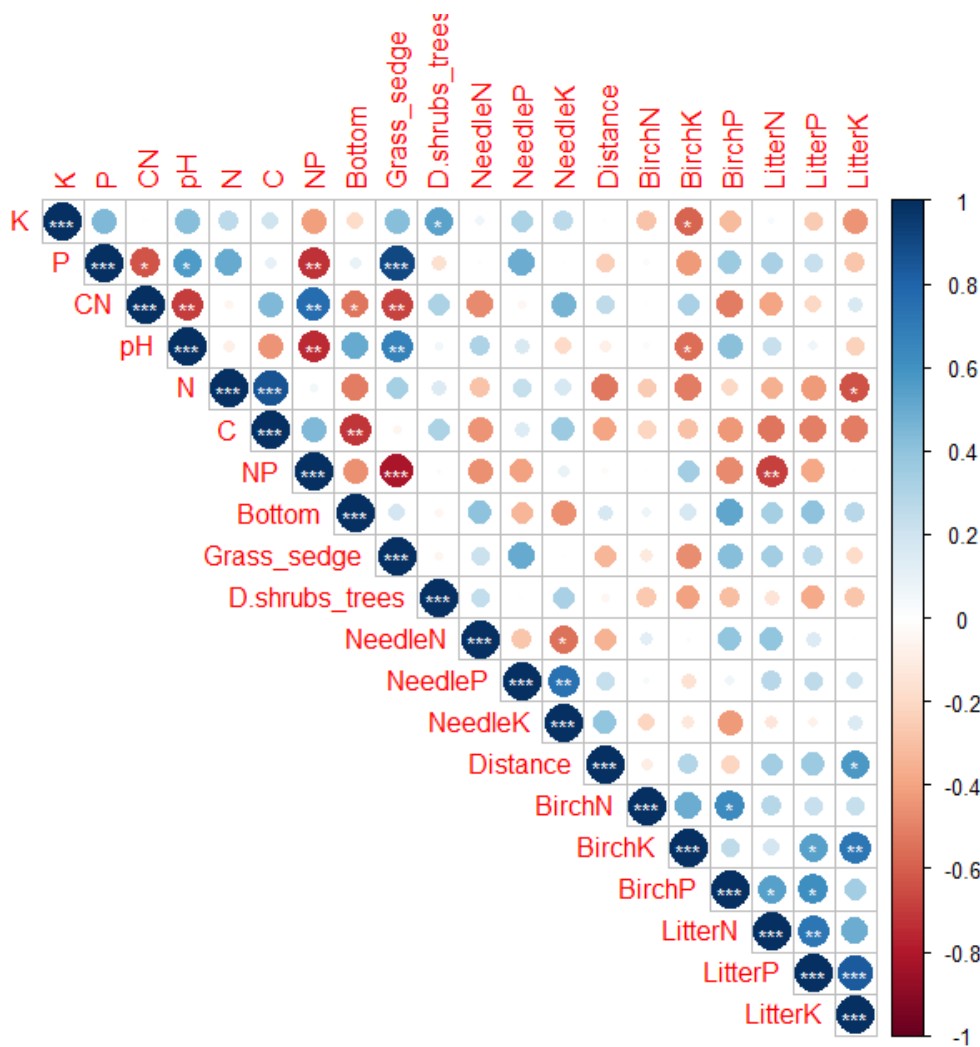

5    **Figure 5. The correlation figure including soil elements (K, P, C:N, pH, C, N:P), the number of species in different layers (the bottom layer, grasses, herbs and sedges, dwarf shrubs and trees), needle elements (N, P, K), plot distance from Sokli, green birch elements (N, P, K) and birch litter elements (N, P, K). Levels of significance * = 0.05, ** = 0.01, *** = 0.001.**




**Figure 6: The upper panel (a and b) presents the ordination pattern of the research plots in dimensions 1 & 2, and the lower panel (c and d) in dimensions 1 & 3. Figures (a) and (c) give plot ordinations and b and d weighted averages of the most abundant species. Less abundant species are marked with light-coloured crosses. The names of species are combinations of the first four letters of generic and species names (e.g. Solivirg = *Solidago virgaurea*). The tree species mentioned in the fig are at seedling stage. Plots D4 and B3 were located on top of each other and only D4 is shown.**



**Figure 7: Ordination pattern with smooth surface fit and linear vector fit of soil P (a), linear vector fits of soil element contents (b), linear vector fits of foliar data as well as plot distance from Sokli phosphate ore (c), and linear vector fits of number of species in different layers of understory (d). Bottom layer includes moss and lichen species, Grass_sedge includes forb, grass, and sedge species and D.shrubs_trees includes dwarf shrubs and tree seedlings. Plots D4 and B3 were located on top of each other and only D4 is shown.**





**Table 1. Meteorological parameters from Värriö. SWE=snow water equivalent, \*= data from SMEAR 1 station, \*\*= data from SMEAR station, only 2009-2015, otherwise data is collected from Värriö Reseach Station by Finnish Meteorological Institute. The values for climatological normal period are from Pirinen et al. 2012.**

|  | 2014 | 2015 | Climatological normal period (1981-2010) |
|---|---|---|---|
| Mean annual temperature (℃) | 0.84 | 0.95 | -0.5 |
| Degree days | 860 | 640 | 680 |
| Total precipitation (mm) | 610 | 660 | 601 |
| · Snowfall (mm, SWE)* | 390 | 420 | 400 ** |
| · Rainfall (mm) | 220 | 240 | 190 ** |



5   **Table 2. Mean foliar element contents of the three major nutrients and C, and the relationships of C:N and N:P with standard deviations. Units for elements are mg kg⁻¹ for K and P, and g kg⁻¹ for N and C.**

|  | K | P | N | C | C:N | N:P |
|---|---|---|---|---|---|---|
| Pine C | 4480 (910.0) | 1420 (140.0) | 14.10 (0.80) | 510.0 (3.70) | 36.0 (1.90) | 10.0 (1.10) |
| Pine C+1 | 3650 (370.0) | 1150 (76.20) | 13.80 (0.90) | 510.0 (7.30) | 38.0 (2.40) | 11.80 (1.10) |
| Pine C+2 | 3520 (330.0) | 1160 (89.0) | 12.10 (3.70) | 480.0 (140.0) | 36.6 (11.20) | 10.60 (3.30) |
| Spruce C | 6430 (870.0) | 1700 (190) | 12.0 (1.00) | 500 (3.80) | 42.0 (3.50) | 7.10 (0.70) |
| Spruce C+1 | 4240 (850.0) | 1470 (210.0) | 10.70 (4.20) | 440.0 (170.0) | 37.0 (14.50) | 7.30 (2.90) |
| Spruce C+2 | 3650 (750.0) | 1360 (230.0) | 10.10 (3.90) | 440.0 (165.0) | 39.2 (14.90) | 7.50 (3.00) |
| Birch, green | 8170.0 (1450.0) | 2580 (340.0) | 25.0 (1.20) | 470.0 (3.40) | 19.0 (1.0) | 9.80 (1.0) |
| Birch, litter | 2380.0 (1010.0) | 1280.0 (450.0) | 10.10 (1.40) | 490.0 (6.0) | 50.0 (7.40) | 8.50 (2.10) |



**Table 3. Tree species composition of the research plots**

| Plot | Trees/ha | Total volume of trees (m3/ha) | Tree species composition (% of volume) |
|------|----------|-------------------------------|-----------------------------------------|
| A1 | 1300 | 75 | Pinus sylvestris 58.9, Picea abies 0.1, Betula pubescens 41 |
| A2 | 1200 | 78 | Pinus sylvestris 88, Picea abies 3, Betula pubescens 9 |
| A3 | 900 | 46 | Picea abies 1, Betula pubescens 99 |
| A4 | 600 | 130 | Pinus sylvestris 66, Picea abies 25, Betula pubescens 9 |
| A5 | 1200 | 125 | Pinus sylvestris 77, Picea abies 15, Betula pubescens 8 |
| A6 | 500 | 54 | Pinus sylvestris 99, Betula pubescens 1 |
| B1 | 300 | 3 | Pinus sylvestris 93, Picea abies 1, Betula pubescens 6 |
| B2 | 300 | 33 | Pinus sylvestris 100 |
| B3 | 500 | 128 | Pinus sylvestris 99.5, Picea abies 0.2, Betula pubescens 0.3 |
| C1 | 800 | 14 | Pinus sylvestris 100 |
| C2 | 1100 | 105 | Pinus sylvestris 48, Picea abies 34, Betula pubescens 18 |
| D1 | 700 | 99 | Pinus sylvestris 99.9, Betula pubescens 0.1 |
| D2 | 1100 | 48 | Betula pubescens 100 |
| D3 | 500 | 18 | Pinus sylvestris 86, Betula pubescens 14 |
| D4 | 300 | 43 | Pinus sylvestris 34, Picea abies 37 , Betula pubescens 29 |
| D5 | 700 | 98 | Pinus sylvestris 100 |



**Table 4. Linear correlations of element contents of soil, needles and leaves, number of species in different vegetation layers and plot distance from Sokli with the NMDS ordination pattern. Bottom layer includes moss and lichen species, grasses and sedges includes forb, grass and sedge species and d. shrubs and trees include dwarf shrubs and tree seedlings. Levels of significance: º = 0.1, * = 0.05, ** = 0.01, *** = 0.001.**

| Variable | $R^2$ | $p <$ |
|---|---|---|
| K | 0.199 | 0.287 |
| P | 0.726 | 0.001*** |
| N | 0.414 | 0.050* |
| C | 0.171 | 0.347 |
| C:N | 0.606 | 0.005** |
| N:P | 0.380 | 0.064º |
| pH | 0.316 | 0.126 |
| Needle P | 0.235 | 0.243 |
| Needle N | 0.013 | 0.932 |
| Needle K | 0.361 | 0.085º |
| Birch P | 0.410 | 0.062º |
| Birch N | 0.179 | 0.324 |
| Birch K | 0.303 | 0.121 |
| Bottom layer, species number | 0.148 | 0.408 |
| Grasses and sedges, species number | 0.721 | 0.002** |
| D. shrubs and trees, species number | 0.375 | 0.070º |
| Plot distance from Sokli | 0.214 | 0.269 |





## Appendix A. Soil element contents across plots

**Table A1. Mean total contents of elements in soil layers and their standard deviations in parenthesis. All plots are included. Units**
5 **for K and P are mg kg$^{-1}$, and for N and C g kg$^{-1}$.**

|         | K        | P          | N         | C        | C:N       | N:P        | pH        |
|---------|----------|------------|-----------|----------|-----------|------------|-----------|
| F layer | 830 (300) | 810 (320) | 13.3 (2.9) | 420 (98) | 32 (6.8) | 17.8 (5.5) | -         |
| O layer | 490 (220) | 720 (500) | 9.9 (3.7) | 260 (106) | 26 (5.7) | 17.2 (7.3) | 3.7 (0.2) |
| A layer | 320 (220) | 380 (510) | -         | -        | -         | -          | -         |
| B layer | 590 (230) | 1030 (1300) | -       | -        | -         | -          | -         |



**Table A2. Plots differing from other plots in soil total P, p< 0.05. Letters F, O, A and B denote soil horizons.**

| All plots | A1 | A3 | A4 | B1 | D2 |
|---|---|---|---|---|---|
| A1 | - | O, A, B | - | A, B | - |
| A2 | - | All layers | - | All layers | O |
| A3 | O, A, B | - | All layers | O, B | O, A, B |
| A4 | O | All layers | - | All layers | O |
| A5 | B | All layers | - | All layers | O |
| A6 | B | All layers | B | All layers | O |
| B1 | A, B | O, B | All layers | - | A, B |
| B2 | - | All layers | - | F, A, B | - |
| B3 | - | All layers | - | All layers | O |
| C1 | F | All layers | - | All layers | F, O |
| C2 | B | All layers | B | F, A, B | - |
| D1 | B | All layers | - | All layers | O |
| D2 | - | O, A, B | F, O | A, B | - |
| D3 | F, B | All layers | B | F, A, B | F |
| D4 | B | All layers | B | All layers | F, O |
| D5 | F, B | All layers | B | All layers | F, O |





**Table A3. Plots differing from other plots in soil total N, p<0.05. Letters F and O denote soil horizons.**

| All plots | A3 | A6 | C2 | D2 | D3 |
|---|---|---|---|---|---|
| A1 | - | F | - | - | - |
| A2 | F | F | - | - | - |
| A3 | - | F, O | - | - | - |
| A4 | F, O | F | O | F,O | O |
| A5 | F, O | F | O | O | O |
| A6 | F, O | F | F, O | F,O | F |
| B1 | F | - | - | F, O | O |
| B2 | - | F | - | - | - |
| B3 | - | F | - | O | O |
| C1 | F | - | - | F | O |
| C2 | - | F, O | - | - | O |
| D1 | F | F | - | F | O |
| D2 | - | F, O | - | - | - |
| D3 | - | F | O | - | O |
| D4 | F | F | - | F | O |
| D5 | F, O | F | O | F,O | O |



**Table A4. Plots differing from other plots in soil C:N ratio, p<0.05. Letters F and O denote soil horizons.**

| All plots | A6 | B1 | B2 | C2 | D3 |
|---|---|---|---|---|---|
| A1 | F | F | F | - | - |
| A2 | F | F | F | - | - |
| A3 | F | F | F | - | - |
| A4 | F | F | F | O | - |
| A5 | F | F | F | O | O |
| A6 | - | - | - | F, O | F, O |
| B1 | - | - | - | F, O | F, O |
| B2 | - | - | - | F, O | F, O |
| B3 | F | F | F | - | - |
| C1 | F | F | F | - | - |
| C2 | F, O | F, O | F, O | - | - |
| D1 | F | F | F | - | - |
| D2 | F | F | F | - | - |
| D3 | F, O | F | F, O | - | - |
| D4 | F | F | F | - | F |
| D5 | F | - | F | - | O |




**Appendix B: Needle and leaf nutrient contents per plot**

**Table B1. Statistically significant differences between different needle and leaf age groups (C = youngest needles , C+1 = one-year-old needles, C+2 = two-year-old needles), p<0.05**

|  | P | N | C | C:N | N:P |
|---|---|---|---|---|---|
| Pine | C & C+1, C & C+2 | C & C+2 | C & C+1, C & C+2 | C & C+2 | C & C+1, C & C+2 |
| Spruce | C & C+1, C & C+2 | No differences between age classes | No differences between age classes | No differences between age classes | C & C+1, C & C+2 |



**Table B2. Statistically significant differences of needle nutrient contents between plots, p<0.05.**

|        | P | N | C | C:N | N:P |
|--------|---|---|---|-----|-----|
| Pine | No differences between plots | Plots: 1&6, 2&11, 5&11, 6&11, 7&11, 6&14, 7&14 | No differences between plots | Plots: 11&5, 11&6, 11&7 | No differences between plots |
| Spruce | Plots: 10&4 | Plots: 3&1, 4&1, 8&1, 8&2, 8&6, 10&8, 12&8, 15&8 | Plots: 2&1, 3&1, 6&1, 15&1, 4&2, 12&2, 12&3, 6&4, 15&4, 12&6 | Plots: 3&1, 4&1, 8&1, 8&2, 8&6, 12&8, 15&8 | No differences between plots |



**Appendix C: Average coverage (%) of understory plant species per plot**

| Species | A1 | A1 | A3 | A4 | A5 | A6 | B1 | B2 | B3 | C1 | C2 | D1 | D2 | D3 | D4 | D5 |
|---|---|---|---|---|---|---|---|---|---|---|---|---|---|---|---|---|
| *Pleurozium schreberi* | 43.3 | 51.3 | 39.6 | 59.3 | 44.4 | 57.1 | 25.0 | 24.7 | 64.7 | 9.0 | 38.8 | 0.3 | 1.2 | 53.5 | 14.5 | 38.7 |
| *Hylocomium splendens* | 38.7 | 8.3 | 22.8 | 9.6 | 3.3 | 1.7 | - | 3.8 | 1.3 | - | 28.9 | - | 44.6 | - | 14.5 | - |
| *Dicranum scoparium* | - | 12.6 | - | 4.2 | 5.9 | 10.7 | 1.6 | 2.8 | - | 8.3 | - | 72.9 | - | 1.0 | 9.7 | 4.1 |
| *Dicranum polysetum* | - | - | - | - | 0.2 | 0.1 | - | 0.9 | 0.8 | - | - | - | - | - | 0.6 | 0.4 |
| *Dicranum majus* | - | - | - | 1.1 | - | - | - | - | - | - | - | - | - | - | - | 1.3 |
| *Barbilophozia barbata* | - | - | - | 0.1 | 0.7 | 0.6 | - | - | - | - | - | - | - | 0.2 | 0.2 | 5.8 |
| *Polytrichum strictum* | - | - | 2.8 | - | - | - | 5.0 | 5.3 | - | 0.8 | - | - | - | 2.1 | - | - |
| *Polytrichum commune* | 3.4 | 0.4 | 0.7 | 0.1 | - | - | 0.4 | - | - | - | 1.4 | - | 23.4 | 4.0 | 4.9 | 0.1 |
| *Aulacomnium palustre* | - | - | - | - | - | - | - | - | - | - | - | - | - | 3.7 | - | - |
| *Sphagnum angustifolium* | - | - | - | - | - | - | - | - | - | - | 1.0 | - | - | - | - | - |
| *Sphagnum girgensohnii* | - | - | - | - | - | - | - | - | - | - | 0.3 | - | - | - | - | - |
| *Sphagnum capillifolium* | - | - | - | - | - | - | - | - | - | - | - | - | - | 0.2 | - | - |
| *Ptilidium ciliare* | - | - | - | - | - | 1.1 | - | - | - | - | - | - | - | - | - | - |
| *Peltigera aphthosa* | 1.1 | - | - | - | 1.0 | - | 0.6 | 0.7 | 0.4 | - | - | - | - | - | 0.1 | - |
| *Peltigera rufescens* | - | - | 0.2 | - | - | - | - | - | - | - | - | - | - | - | - | - |
| *Peltigera neopolydactyla* | - | - | - | - | - | - | 10.9 | 0.1 | - | - | - | - | - | - | - | - |
| *Nephroma arcticum* | - | 2.3 | 1.0 | 3.9 | - | - | 6.1 | 1.5 | 0.5 | - | - | - | - | 0.5 | 0.1 | 1.3 |
| *Umbilicaria deusta* | - | - | - | - | - | - | - | 0.1 | - | - | - | - | - | - | - | - |
| *Cladonia rangiferina* | 0.3 | 1.0 | 0.3 | 1.6 | 1.5 | 2.0 | 6.2 | 2.8 | 1.6 | 31.0 | - | 3.8 | - | 5.2 | 0.3 | 5.3 |
| *Cladonia cornuta* | - | - | 0.1 | - | 0.2 | - | 0.1 | 0.2 | - | 0.1 | - | 0.2 | - | - | - | 0.3 |
| *Cladonia stellaris* | - | - | - | - | - | - | - | - | - | - | - | - | - | - | - | - |
| *Cladonia deformis* | - | - | - | - | - | - | - | 0.1 | - | 0.6 | - | - | - | - | - | 0.1 |
| *Cladonia crispata var. crispata* | - | - | - | - | - | - | - | - | - | - | - | - | - | - | - | - |
| *Cetraria islandica* | - | 0.1 | - | - | - | - | - | - | - | - | - | - | - | - | - | - |
| *Stereocaulon tomentosum* | - | - | - | - | - | - | 4.7 | 2.2 | - | - | - | - | - | - | - | - |
| *Linnaea borealis* | - | - | - | - | 0.3 | 0.2 | - | - | 0.1 | - | - | - | - | - | - | - |
| *Vaccinium myrtillus* | 1.2 | 4.3 | - | 10.7 | 14.7 | 27.7 | - | 5.2 | 2.3 | 1.4 | 8.4 | 1.3 | 2.7 | 18.8 | 6.0 | 6.9 |
| *Vaccinium vitis-idaea* | 15.3 | 26.8 | 5.6 | 22.2 | 11.9 | 5.2 | 1.3 | 5.5 | 21.6 | 5.7 | 3.4 | 28.2 | 25.8 | 4.0 | 7.4 | 4.4 |




| | | | | | | | | | | | | | | | | |
|---|---|---|---|---|---|---|---|---|---|---|---|---|---|---|---|---|
| *Vaccinium uliginosum* | 7.1 | 0.8 | - | - | - | 0.5 | 15.5 | 5.3 | 2.3 | 2.9 | 34.3 | - | 11.6 | 41.7 | 6.3 | 1.5 |
| *Empetrum nigrum* | 2.7 | 9.8 | 0.8 | 8.4 | 14.1 | 12.4 | - | 9.6 | 32.8 | 9.3 | 14.5 | 15.0 | 4.1 | 40.4 | 7.0 | 6.9 |
| *Arctostaphylos uva-ursi* | - | - | - | - | - | - | 10.7 | 0.2 | 0.4 | - | - | - | - | - | - | - |
| *Arctostaphylos alpina* | - | - | - | - | - | - | - | - | - | - | - | - | - | - | - | - |
| *Betula nana* | 23.9 | 6.3 | 1.3 | - | - | - | - | - | - | - | 2.2 | - | - | 2.7 | - | - |
| *Calluna vulgaris* | - | - | - | - | - | 0.5 | - | - | - | 0.2 | - | - | - | 0.6 | - | 0.4 |
| *Rhododendron tomentosum* | - | - | - | - | - | - | - | - | - | - | - | - | - | 0.8 | 3.6 | - |
| *Juniperus communis* | 0.9 | - | 0.8 | 0.6 | - | - | 5.7 | 0.3 | 2.2 | - | 2.5 | - | 3.7 | - | - | - |
| *Picea abies* | - | - | - | 0.5 | - | - | - | - | - | - | - | - | - | - | - | - |
| *Pinus sylvestris* | - | - | - | - | - | - | 1.6 | 0.8 | - | 2.1 | 0.1 | 0.9 | - | 0.8 | - | 0.7 |
| *Betula pubescens* | 1.1 | 0.2 | 1.4 | - | - | 0.4 | 2.5 | 0.7 | - | - | - | - | - | 0.2 | 0.1 | - |
| *Populus tremula* | - | - | - | - | - | 0.4 | - | - | - | - | - | - | - | - | - | - |
| *Diphasiastrum complanatum* | - | - | - | - | - | 0.1 | 0.7 | - | - | - | - | - | - | - | - | - |
| *Trientalis europaea* | - | 0.1 | 0.7 | - | - | - | - | - | - | - | - | - | 3.4 | - | - | - |
| *Melampyrum sylvaticum* | - | - | 0.3 | - | - | - | 0.1 | - | - | - | - | - | - | 0.1 | - | - |
| *Solidago virgaurea* | - | - | 1.1 | - | - | - | 1.2 | - | - | - | 0.2 | - | 0.7 | - | - | - |
| *Rubus arcticus* | - | - | 0.6 | - | - | - | - | - | - | - | - | - | 3.2 | - | - | - |
| *Rubus chamaemorus* | - | - | - | - | - | - | - | - | - | - | 0.4 | - | - | - | - | - |
| *Antennaria dioica* | - | - | - | - | - | - | 1.2 | - | - | - | - | - | - | - | - | - |
| *Orthilia secunda* | - | - | - | - | - | - | - | - | - | - | 0.1 | - | - | - | - | - |
| *Epilobium angustifolium* | - | - | - | - | - | - | - | - | - | - | - | - | 0.7 | - | - | - |
| *Galium uliginosum* | - | - | - | - | - | - | - | - | - | - | - | - | 0.1 | - | - | - |
| *Geranium sylvaticum* | - | - | - | - | - | - | - | - | - | - | - | - | 0.2 | - | - | - |
| *Chelidonium majus* | - | - | - | - | - | - | - | - | - | - | - | - | 0.1 | - | - | - |
| *Comarum palustre* | - | - | - | - | - | - | - | - | - | - | - | - | 0.3 | - | - | - |
| *Equisetum sylvaticum* | 0.1 | - | - | - | - | - | - | - | - | - | - | - | 0.8 | - | 0.2 | - |
| *Luzula pilosa* | 0.6 | - | 0.4 | - | - | - | 1.8 | - | - | - | 0.2 | - | 2.4 | 0.3 | - | - |
| *Elymus repens* | - | - | - | - | - | - | - | - | - | - | - | - | - | - | - | - |
| *Deschampsia flexuosa* | 3.5 | 0.6 | 10.3 | 0.6 | - | 0.2 | 2.1 | 0.3 | 0.4 | - | 2.8 | - | 15.9 | 7.1 | 2.2 | 0.2 |
| *Festuca rubra* | - | - | - | - | - | - | - | - | - | - | - | - | 1.9 | - | - | - |
| *Calamagrostis epigejos* | - | - | - | - | - | - | - | - | - | - | - | - | 5.4 | - | - | - |
| *Carex digitata* | - | - | - | - | - | - | - | - | - | - | - | - | 0.6 | - | - | - |



| | | | | | | | | | | | | | | | | |
|---|---|---|---|---|---|---|---|---|---|---|---|---|---|---|---|---|
| *Carex nigra* | - | - | - | - | - | - | - | 0.2 | - | - | - | - | - | - | - | - |
| *Carex canescens* | - | - | - | - | - | - | - | - | - | - | 0.5 | - | 3.4 | - | - | - |
| *Carex globularis* | - | - | - | - | - | - | - | - | - | - | - | - | - | 1.7 | - | - |