# Peer review of "Soil total phosphorus and nitrogen explain vegetation community composition in a northern forest ecosystem near a phosphate massif"

_Biogeosciences, 2019_

## Referee Comment (RC1) · Anonymous Referee #1 · 15 Aug 2019

Soil nutrients and stoichiometry is an important topic in forest ecosystems. The manuscript studied the relationships between understory vegetation species abundance in a boreal forest and soil /leaf nutrients. However, there are several major concerns about the statistical methods, the data presented in the figures and tables and shortage of basic information regarding the study site. Additionally, there are also grammatical issues and inappropriate descriptions of the results. The discussion cannot fully support their hypotheses and results.

Statistical analysis: One-way ANOVA were chosen in the manuscript, it implies that the plot was the only factor. However, tree species plays an important role in soil nutrients

as well, thus tree species should be considered as a confounding factor. Due to high spatial heterogeneity in soil samples, when you determine the difference in different plots, the block effects also should be considered in the statistical analysis;

2) Considering the relationships between species composition and soil nutrients, besides the species, the authors also should treat age classes as the second factor. For the same reason, soil layer also should be considered in the analysis.

shortage of basic information: the authors provided the basic tree and other species composition in the Table 3. The tree age and biomass also affects the soil nutrients. The author should provide the mean basal area, leaf area index and mean DBH. These basic information would be useful to estimate the effects of tree species on the understory species composition and on soil nutrients. As to the weather information, the min and max temperature should also be provided in Table 1.

Problems with presentation the data in figures and tables. 1) Authors should add a new table/Figure to show the mean soil nutrients in the birch, scots pine and spruce plots in each layer and make stat analysis.

2) In Fig 5, there were no adj-R2 value to show which factor possessed the most weight. At the same time, these correlations could be better presented in Table not in fig.

3)the confusing plot numbers in table A2/A3 /A4 and Table B2. In the Table A, the plot number was in alphabetical order while the Arabic number was adopted in table B2.

4) we cannot find the stats evidence support the data. For example: "Foliar N:P ratio did not show any differences in either species between plots.. . . .. . . ..green leaves compared to other species." (3.2)

Writing issues:

1) In the results section, the first sentence in each sub section provides meaningless information for the data and these sentences can be deleted. For example: " The average contents. . .. . .. In fig 4" (3.1 soil element contents). The same was also found

in each paragraph.

2) There were some grammatical issues in each paragraph. There was no deep discussion to support the hypotheses and results.

---

## Referee Comment (RC2) · Anonymous Referee #2 · 26 Aug 2019

**General comments**

Matkala et al propose a study of the relationships between vegetation community structure and leaf elemental composition, and environmental factors in a northern forest ecosystem of Finland. In line with the recent literature raising the fact that P availability can also be a strong driver of vegetation growth in northern ecosystems, the authors put a stress on total N and P contents in humus and soil layers. They justify this study by the need to establish a baseline status of the ecosystem before potential disturbance that may arise due to possible P mining in the region. It seems the authors want to assess if understory vegetation community composition can predict the ecosystem

nutrient status to help developing a rather low cost monitoring but this is not developed further in the discussion.

The authors collected a substantial amount of data that should contribute to address their objectives. However, I have several concerns about the formulation of objectives, result analysis and discussion. In particular, both the text and the presentation of the results could be shortened and simplified, notably by focusing on clear objectives. This would help the reader to faster understand what the authors did exactly, and the reasoning leading to their conclusions.

To my opinion, this study has the potential to address important questions that falls well within the scope of *Biogeosciences*; however, the present manuscript requires a thorough revision before publication.

**Specific comments**

Title: rephrase; "understory vegetation" is not precise, that are the species composition and abundance that were studied. Why to focus on the understory since the tree cover was also studied? You could also focus on what you consider as your main result, for example "Soil total P explains vegetation community composition in a northern boreal forest ecosystem".

Abstract: needs a sentence on sampling design (the way the relationship were addressed)... In the present case, the reader have no idea what the "plots" refer to and what to conclude from that information. Here and in the materials and methods section, you need to state clearly that you described vegetation, and sampled tree leaves and soil, at different distances from the P ore. Revise this abstract after clarifying the objectives and re-analyzing/discussing the results.

Introduction: could be simplified and shortened. Also needs to better formulate the objective(s) and hypotheses, and/or to provide all the information that lead to such hypotheses. In particular, it is not very clear why the hypotheses focus on the humus layer.

By comparing the title, abstract and introduction, it is not clear if the objective if finally (1) to explain the understory species composition and abundance with environmental parameters, and particularly soil total P content, or (2) to predict soil/environment nutrient status by surveying understory vegetation. One option should be chosen and the whole article built around.

Material and methods: The site selection process is not clear; in particular, what is the basis for selecting those transects? I did not get if there is any gradient, for example. Are all the study sites located at a similar elevation with similar climate conditions? As the study sites were located on different geological units (Fig. 1), did the authors tried to include such factor in their analysis? Do we know anything about the P contents of those rocks? Are these rocks essential parent material for the soils developed at the sites? Any idea of the age/development stage of the soils? Are all the soils studied haplic podzols? As for the statistical analyses, it seems the forest stand composition could be better taken into account by accounting for the species % of volume or by grouping sites according to their dominant species. As raised by reviewer 1, stand age could also be a confounding factor. I think the authors could try to better explain the variations they observe in soil values. Also, why did the authors focused in the elemental contents of the O horizon (humus) in their analysis? Do we have an idea of the distribution of fine roots (and vegetation uptake zone) in the soil profile? The use of understory species is interesting, would it be possible to go further by narrowing down the number of species, by detecting indicator species (of the P status for example), and building a "simple" prediction model?

Results: I feel some results are not presented in the way that best help to address the questions of interest. I think in particular about the Fig. 3 and 4 or Table A2–A4, where we don't have any clue about what could lead the variability and differences (forest stand composition? rock parent material? other?).

Discussion: The authors have put honorable efforts in comparing the data they obtained with known ranges of values for similar areas published in the literature. However, this part of the discussion could be better synthesized and written in a simpler and shorter style. The discussion lacks development on the results obtained in regard to the objectives of the study. What is the functional significance of these results? When focusing on key results, the Fig. and/or Tables where they are presented should be reminded to help the reader.

Conclusions: potentially revise according to modifications in the introduction and discussion.

**Technical corrections**

Whole text: refer to Fig., Tables, and Appendix when appropriated, and order and number them following their first apparition in the text.

p. 1, l. 8: "We studied the relationship of forest understory vegetation with nutrient contents of soil and tree leaves...": write something which fits with your objectives and title. Again, "understory vegetation"is not precise, add "species composition and abundance", which appears only at l. 12–13.

p. 1, l. 9–10: add a comma: "At most study plots**,** boreal..."

p. 1, l. 11–12: here and elsewhere, change "abundance and species composition of the vegetation" to "species composition and abundance of the understory vegetation".

p. 1, l. 13–14: what is the information you want to raise?

p. 1, l. 19: some fixes: "... controlling the **species** composition**s** of tree **stand** and understory..."

p. 1, l. 21: what do you mean by "modified"? I would rather say that those ecosystems are "characteristically cold, have a short growing season, and are nutrient-poor".

p. 1, l. 21–22: "affects" is not precise, and the sentence could be shorter and simpler. Suggestion: "Organic matter decomposition and nutrient release are usually slow in cold climates".

p. 1, l. 22–24: not very informative.

p. 1, l. 25: change "tree species affect" by "tree cover affects", unless you want to precise "different tree species affect differently understory..." (if this fits to the references cited).

p. 1, l. 26: again, "understory vegetation" is not precise, which parameters? Focus on what you study here (i.e. species composition and abundance).

p. 1, l. 27: "litterfall" is sufficient since it also includes branches, etc.

p. 1, l. 29: N and P "are generally the **main** growth-limiting nutrients..."

p. 1, l. 29–p. 2, l. 16: this whole paragraph convey interesting information but is not enough focused for the present study. It can be shortened and simplified by synthesizing the main ideas.

p. 2, l. 3–4: useless information. In the context of vegetation growth, available N mostly derives from organic matter decomposition (unless the plant is a N-fixer), and available P both from weathering and organic matter decomposition.

p. 2, l. 4–6: not necessary in the context of this article.

p. 2, l. 8–9: N–P interaction is a bit cryptic (is that a statistical term?), can you say something more functional? I think the idea is that the coupling between the N and P cycles drives nutrient limitation.

p. 2, l. 11: move the comma: "In boreal N-limited forests**,** ..."

p. 2, l. 18: replace "soil nutrients" by "soil nutrient content" or "soil total N and P".

p. 2, l. 27–31: this is not related to your study and could be considered as a confounding factor hindering potential interesting relations. Move the information to the material and methods, and state clearly that you assume reindeer pressure (grazing, trampling, but also nutrient exports or inputs) is not such a confounding factor for this study. Of course,

it has to be the case! Did you evaluate somehow the reindeer pressure at your study sites? How? Was it important? Was it typical for the region? Was it constant across sites?

p. 2, l. 32: what do you mean by "undisturbed"? The current (steady) state? A baseline status?

End of the introduction: rephrase and clarify objective(s).

p. 3, l. 1: move this (= method) after the effects of mining.

p. 3, l. 3–4: too much! Focus on mining and keep reindeer and/or climate change for an opening in the discussion. Other option: start the last paragraph of the intro saying that several disturbances such as mining, grazing (is there a change in grazing in the region? why?), and climate change could affect nutrient status of the ecosystem and you aim at establishing a baseline to monitor the effects of those disturbances. In that case, keep something on reindeer and/or add something on climate change, but keep it short and focused (effects on nutrient status of ecosystem).

p. 3, l. 4–5: simplify and shorten! It is not very clear why you did this... Did you want to establish a relatively simple and cheap protocol of monitoring? For example, surveys of key understory species abundance that would be indicative of the ecosystem nutrient status?

p. 3, l. 6–7: this sentence does not justify these hypotheses. Delete and replace by something like "We hypothesized: a)... b)...", and move that at the end of the preceding paragraph.

p. 3, l. 9–11: are these hypotheses justified with the preceding text of the introduction? Do they really relate to the objective? Why to focus suddenly on "humus"? I feel some pieces of the reasoning are missing.

p. 3, l. 11: wouldn't it be rather the humus layer that reflects the nutrient content of the leaves? Unless you assume most of the tree uptake occurs in this layer.

p. 3, l. 14: what are these transects? Are they organized along a gradient? Which one? It seems on the map that it would be the distance to the carbonatite massif.

p. 3, l. 15: "No plots were located inside the mining district", why?

p. 3, l. 19: needs reference, but isn't that what you want to study? Consider moving this info to the discussion.

p. 3, l. 22: start the sentence with "Thus,"

p. 3, l. 23–24: delete ", but they were not on any Natura area".

p. 3, l. 26: what is the gradient?

p. 3, l. 32: add a comma after "5°C".

p. 4, l. 1: change subsection title to "Plot setup and vegetation characterization".

p. 4, l. 5: change subsection title to "Sampling of soil".

p. 4, l. 6–10: move this to the preceding section.

p. 4, l. 6: cite Table 3 and add to the table tree height and diameter info.

p. 4, l. 7: precise "cover (% **surface area**)".

p. 4, l. 8: cite appendix C.

p. 4, l. 10: add comma after "Altogether".

p. 4, l. 11: provide diameter of the soil corer.

p. 4, l. 11: change to "The soil was sampled within one meter from the subplots".

p. 4, l. 12: change "The samples" to "The soil cores".

p. 4, l. 14–15: simplify and shorten.

p. 4, l. 16: remove "already".

p. 4, l. 18: remove "samples from".

p. 4, l. 20–21: two first sentences useless, delete and add "2015" after "September" in the third sentence.

p. 4, l. 30: remove ", totalling of 100 leaves per plot".

p. 5, l. 2: replace "in a similar way than needles" by "at 65°C for 48 h".

p. 5, l. 2: concretely, how did you clean the leaves? With a brush? Deionized water? Other?

p. 5, l. 11: fix: "two to three mg of sample **were**..."

p. 5, l. 11: "VarioMax analyser" is a machine, what is the method behind?

p. 5, l. 12: idem, replace "MilliQ water" by "ultrapure water".

p. 5, l. 21: "Ordination pattern of the study plots and weighted averages of plant species", not clear, I thought you ordinated the weighted averages.

p. 5, l. 23: state why you did not present results for dim 2 vs dim 3.

p. 5, l. 24: "some other environmental variables", list them.

p. 6, l. 5: "... imply rather high variation between and within plots", isn't that what you want to study? Maybe a deeper analysis, including tree species, forest stand age, and/or geology could help explaining a bit the variability observed.

p. 6, l. 5–6: "Other soil elements...", precise which ones or delete the sentence.

p. 6, l. 11–12: the higher P content in young needles may indicate reallocation processes, which could be discussed shortly in regard to P availability for example.

p. 6, l. 12: replace "Unlike the expectations" by "Unlike **our** expectations".

p. 6, l. 15–23: refer to the Table or Fig. these informations are presented (if you go back to Table 2 after referring to Table B2, for example).

p. 6, l. 19: replace "discovered" by "detected".

p. 6, l. 22: "Çÿumber of species", is this a good variable for your objectives? Could the total % cover of different species groups (the ones of Fig. 5, for example) be more informative?

p. 6, l. 27: how do you define the left and right sides of the plots? Is there a threshold or is this empirical?

p. 6, l. 28: do you mean Fig. 6b? Or 6a–b?

p. 6, l. 30: no need to cite Table 3 here.

p. 7, l. 10: Start the paragraph by citing the Fig.: "In Fig. 7c–d, ..."

p. 7, l. 18: did you analyze soil samples by plot or by cluster? Did you quantify both within- and between-plot variabilities?

p. 7, l. 19: refer to Fig. 3 where the soil P contents are presented. Also, this is a huge variability: is mainly due to between- or within-plot variation? If this is between plots, you might be able to explain it somehow by additional exploration of environmental factors, but if it is within plot there is no hope tree species will help, for example...

p. 7, l. 24: "implying that decaying plant parts were a major source of P", for what? The soil organic layers or plants?

p. 7, l. 24–27: would it be possible to find a pattern of P content in the deep soil layers according to the soil rock parent material/geology? Do you think that high P content in the humus layer is important for plant nutrition (recycling) or is that just high litter production coupled with slow decomposition rates?

p. 7, l. 28: what is the context of the study by Köster et al (2014)?

p. 8, l. 2: and so? Can you say something concrete for your study area?

p. 8, l. 4: replace "similar than" by "similar **to**".

p. 8, l. 18–19: Which analysis/which Fig. or Table? Table 4? But is it species richness or ordination pattern which was regressed? What do you mean exactly by "species richness"? The number of species?

p. 8, l. 19–20 and l. 30: which soil layer(s) are you considering? The hypotheses were about humus.

p. 8, l. 32–p. 9, l. 2: "needles were sampled at different time of year than soil...", this should be mentioned in, and even might only be part of, the materials and method section. If you sampled the needle at a right moment, it should be quite integrative of nutrient availability across the growing season.

p. 9, l. 4–8: why not such an opening but right now it is not well connected to what precedes. You could also talk about the coupling between N and P cycles and how this could be affected by climate change or disturbances and in turn affect ecosystem status and processes.

p. 9, l. 6: "variation in the vegetation", be more precise (which parameter, sense of variation).

p. 9, l. 10 and 17: "vegetation dynamics", this is not what you study here, change to "vegetation community composition" or something like that.

p. 9, l. 12: change "has been discovered" to "was found".

Fig. 2: draw or remind in the title where tree cover was described.

Fig. 3: would it be possible to also represent tree cover species (or different groups based on dominant species)? or age? or geology? Is this graph representing between-plot variability (i.e. you first calculated the mean for each plot and made the boxplots with those means) or a mix between within- and between-plot variability (i.e. you took all sub-plots values to make the boxplot)? It would be interesting to compare between- and within-plot variabilities.

[Figure]

Fig. 4: same comments as for Fig. 3.

Fig. 5: What is the correlation coefficient calculated? (Pearson? Other?) Precise what is the "bottom layer". Why not to call this layer "moss lichen"? It would be clearer.

Fig. 6: what is the criteria to define "the most abundant species"?

Fig. 6, l. 8: replace "generic" by "genera". Start the last sentence of the figure legend by "In (a),"

Fig. 7: remind which soil layer was considered for this analysis (I am assuming it's O).

Table 1: "degree days", shouldn't that be called "sum of degree days"? What is the unit?

Table 2: the classical ordering would be C, N, P, K, C:N, N:P. For numbers (mean and sd), provide the same number of digits after the dot for each column (one is enough) and align the numbers to the right to ease comparison of lines. Why not to put K in the same unit as the others?

Table 3: did you estimate the whole aerial volume of trees or just the trunk volume? Make three sub-columns for each species abundance. Add in this table tree height, diameter,...

Table 4: remind the first seven lines are soil values.

Tables A2–A4: From which statistical test are these table issued? These tables hardly help to address your objectives.

Table B1: title: change to "Statistically significant differences between needle age group by species". From which test?

Appendix C: precise "(% **of surface area**)".

---

## Author Comment (AC2) · 30 Oct 2019

Response to referee #2

Below are the comments of the referee #2 in black and our responses in blue font and changes to manuscript *italicized and gray*.

We thank anonymous referee #2 for the time they have spent revising our manuscript. We found the comments very helpful and sincerely appreciate all the detailed and concrete suggestions on how to proceed with the manuscript. As both referees brought up points about the role of tree species in soil nutrient content, we will add analyses and discussion about this topic to our manuscript.

Title: rephrase; "understory vegetation" is not precise, that are the species composition and abundance that were studied. Why to focus on the understory since the tree cover was also studied? You could also focus on what you consider as your main result, for example "Soil total P explains vegetation community composition in a northern boreal forest ecosystem".

**Response:** We agree that we have studied more than just the understory vegetation, and will reconsider and rephrase the title from that point of view. We appreciate the suggestion, but feel that the title example by the referee would be too strong of a statement. Our study is more of a case study about a very special area surrounding a phosphate massif and, thus, it would be rather risky to generalize the results to cover also other northern boreal forest ecosystems.

Abstract: needs a sentence on sampling design (the way the relationship were addressed)... In the present case, the reader have no idea what the "plots" refer to and what to conclude from that information. Here and in the materials and methods section, you need to state clearly that you described vegetation, and sampled tree leaves and soil, at different distances from the P ore. Revise this abstract after clarifying the objectives and re-analyzing/discussing the results.

**Response:** Agreed. We have added information about how the study plots and measurements were arranged.

Introduction: could be simplified and shortened. Also needs to better formulate the objective(s) and hypotheses, and/or to provide all the information that lead to such hypotheses. In particular, it is not very clear why the hypotheses focus on the humus layer.

**Response:** Agreed. We have simplified the introduction according to the suggestions. We also added that we focused on the humus layer with understory vegetation, as that is the layer where the roots of the understory vegetation is, and from where they take their nutrients. Additionally, we formulated a third hypothesis regarding the role of dominant tree species on soil nutrients.

By comparing the title, abstract and introduction, it is not clear if the objective if finally (1) to explain the understory species composition and abundance with environmental parameters, and particularly soil total P content, or (2) to predict soil/environment nutrient status by surveying understory vegetation. One option should be chosen and the whole article built around.

**Response:** This is a very good point and helped us to clarify the "common thread" of the manuscript. We originally started with option 1) and it is still valid. We aim to explain the understory species composition and abundance with environmental parameters, and see if soil total P content has an effect on them. In addition, we want to figure out what environmental parameters could explain soil nutrient contents, especially total P.

Material and methods: The site selection process is not clear; in particular, what is the basis for selecting those transects? I did not get if there is any gradient, for example.

**Response:** We have added the following sentences to materials and methods (after "We established 16 study plots…):

*" The plots were arranged into transects so that it would be easier to follow what is the distance of the possible mining effect from the phosphate ore."*

Are all the study sites located at a similar elevation with similar climate conditions?

**Response:** Yes they are.

As the study sites were located on different geological units (Fig. 1), did the authors tried to include such factor in their analysis?

**Response:** We have now included the geological unit to the analyses (see latter comment replies).

Do we know anything about the P contents of those rocks? Are these rocks essential parent material for the soils developed at the sites? Any idea of the age/development stage of the soils?

**Response:** The Sokli phosphate ore has been carefully sampled and studied, but the surrounding area lacks such detailed information. We added to the figure caption of Fig 1. that this map shows the bedrock in the region, which is essential parent material for the soil. The bedrock in Finland is among the oldest in Europe but the soils have been modified by the latest ice age. Sokli is different from the surrounding soils also because it is located in a sheltered depression and has not been affected by the erosion caused by latest ice age.

Are all the soils studied haplic podzols?

**Response:** Yes they are.

As for the statistical analyses, it seems the forest stand composition could be better taken into account by accounting for the species % of volume or by grouping sites according to their dominant species. As raised by reviewer 1, stand age could also be a confounding factor.
I think the authors could try to better explain the variations they observe in soil values. Also, why did the authors focused in the elemental contents of the O horizon (humus) in their analysis? Do we have an idea of the distribution of fine roots (and vegetation uptake zone) in the soil profile?

**Response:** We agree with these comments. As mentioned earlier in this document, we originally chose humus as the roots of understory vegetation are mainly in this layer, but we have now done more stat. analyses including other layers as well.
Unfortunately, we do not have exact knowledge of the age of the trees at each plot, as we did not core the trees. We know the approximate age of trees at plot A6 and have used this as help, when we now estimated the tree ages based on their dbh and put the trees in three different classes (young trees 1-9.9 cm, mid-aged 10-14.9 cm and old > 15 cm). We added information about the tree age classification to section 2.2.1. We tested the effect of dominant tree species, tree age, soil parent material/bedrock type, and soil horizon on soil P, N and C:N with linear mixed-effect models and have added the following description to section 2.4:

*""We tested the effects of environmental variables on soil P, N and C:N with linear mixed-effect models. We used dominant tree species, estimated tree age, rock parent material and soil horizon as fixed effects and plot as random effect. Soil P needed to be log-transformed while for N and C:N the visual inspection of residual plots did not reveal obvious deviations from homoscedasticity or normality. We obtained p-values for the fixed effects by likelihood ratio tests, where the full model with all the fixed effects was tested against model where each fixed effect was removed in turn. We used package lme 4 (Bates et al. 2015) in R programme 3.4.3 (R Development Core Team, 2017) for building the models. Pseudo R2-value for the models were calculated by using package r2glmm (Jaeger 2017)The models took the form:*

$$SC_{P,N,CN} = B_0 + B_{dt} + B_{ta} + B_g + B_h + \in , \tag{1}$$

*where $SC_{P,N,CN}$ is the soil nutrient content (P, N or C:N ratio), $B_0$ denotes a fixed intercept parameter, $B_{dt}$ denotes the fixed unknown parameters associated with the dominant tree species, $B_{ta}$ denotes the fixed unknown parameters associated with the age of the dominant tree species, $B_g$ denotes the fixed unknown parameters associated with the rock parent material, $B_h$ denotes the fixed unknown parameters associated with soil horizon. The random effect $\in$ is assumed to take the form:*

$$\in = \propto_p + u , \tag{2}$$

*where $\propto_p$ denotes the random parameters related with the research plot and u is an unobservable error term. Random-effect parameters and random-error term are assumed to follow normal distributions $\propto_p \sim N(0, \sigma_p^2)$ and $u \sim N(0, \sigma_u^2)$."*

We made a new subsection for the results of mixed effect models under section 3 and added a table, which includes the fixed effects and their Chisq values, p-values and pseudoR2 values. To the appendices we added figures of the fitted vs. residuals, q-q plots and histograms of the residuals and removed the unnecessary tables of the previous one-way ANOVAs.

In order to more precisely study the relationship of tree species and understory vegetation, we added the volume of birch per plot to the ordination and to Fig. 7d. We also added the cover (% of surface area) of species in the same species groups to the ordination and Fig. 7d.

Additionally (based on a suggestion from referee #1), we grouped the plots based on their dominant tree species and calculated the means of soil nutrients and ratios in each soil horizon in pine, birch and spruce plots. We made a table of these and compared the nutrient contents in each soil horizon with one-way ANOVA.

The use of understory species is interesting, would it be possible to go further by narrowing down the number of species, by detecting indicator species (of the P status for example), and building a "simple" prediction model?

**Response:** Indeed, this would be interesting. However, we feel that building such a model would require a lot more study plots and data.

Results: I feel some results are not presented in the way that best help to address the questions of interest. I think in particular about the Fig. 3 and 4 or Table A2–A4, where we don't have any clue about what could lead the variability and differences (forest stand composition? rock parent material? other?).

**Response:** Agreed. We have replaced Fig. 3 and 4. with box plots which show the within plot variation in these soil nutrients. We have also deleted Tables A2-A4, as the new table of fixed effects and figures of residuals and q-q plots (described above) are more useful in this context.

Discussion: The authors have put honorable efforts in comparing the data they obtained with known ranges of values for similar areas published in the literature. However, this part of the discussion could be better synthesized and written in a simpler and shorter style. The discussion lacks development on the results obtained in regard to the objectives of the study. What is the functional significance of these results? When focusing on key results, the Fig. and/or Tables where they are presented should be reminded to help the reader.

**Response:** We thank the referee for this comment. We have revised the discussion section based on the comments from both referees. We synthesized and shortened the part of text where we compare our total nutrient contents to previous studies as well as reorganized the sections so that the main results become clear in the first paragraph of the discussion section. We wrote more about the role of tree species in soil P content and highlight how and why our results are important and relate with the previous findings. We have also paid more attention on referring to the Figs and Tables.

Conclusions: potentially revise according to modifications in the introduction and discussion.

**Response:** We have deleted here the following sentences from conclusions: "The mean P content of our birch litter samples was 1280 mg kg-1 , which is higher than the P contents of C+1 and C+2 needles of pine (1150 and 1160  15 mg kg-1, respectively). Most of the plots with high total P in the humus layer had high total P contents also in the B layer, where the maximum content was as high as 5500 mg kg-1."

Technical corrections
Whole text: refer to Fig., Tables, and Appendix when appropriated, and order and number them following their first apparition in the text.

**Response:** We will do the final check just before the re-submission of the manuscript.

p. 1, l. 8: "We studied the relationship of forest understory vegetation with nutrient contents of soil and tree leaves...": write something which fits with your objectives and title. Again, "understory vegetation"is not precise, add "species composition and abundance", which appears only at l. 12–13.

**Response:** We thank the referee for this comment. We have revised the first sentence of the abstract, although we do feel that it already represents our objectives and title. We added "species composition and abundance" as suggested.

p. 1, l. 9–10: add a comma: "At most study plots, boreal..."

**Response:** Ok, done.

p. 1, l. 11–12: here and elsewhere, change "abundance and species composition of the vegetation" to "species composition and abundance of the understory vegetation".

**Response:** Ok, changed.

p. 1, l. 13–14: what is the information you want to raise?

**Response:** That the area where our study was conducted has very high variability in the contents of those soil nutrients which we studied. We have revised the text in order to clarify this.

p. 1, l. 19: some fixes: "... controlling the species compositions of tree stand and understory..."

**Response:** Agreed and corrected.

p. 1, l. 21: what do you mean by "modified"? I would rather say that those ecosystems are "characteristically cold, have a short growing season, and are nutrient-poor".

**Response:** Agreed and changed.

p. 1, l. 21–22: "affects" is not precise, and the sentence could be shorter and simpler. Suggestion: "Organic matter decomposition and nutrient release are usually slow in cold climates".

**Response:** Agreed and changed

p. 1, l. 22–24: not very informative.

**Response:** Agreed, we deleted this sentence.

p. 1, l. 25: change "tree species affect" by "tree cover affects", unless you want to precise "different tree species affect differently understory..." (if this fits to the references cited).

**Response:** Agreed, and changed to tree cover.

p. 1, l. 26: again, "understory vegetation" is not precise, which parameters? Focus on what you study here (i.e. species composition and abundance).

**Response:** Agreed, we changed "understory vegetation" to "species composition and abundance in the understory"

p. 1, l. 27: "litterfall" is sufficient since it also includes branches, etc.

**Response:** Agreed and deleted the word "leaf".

p. 1, l. 29: N and P "are generally the main growth-limiting nutrients..."

**Response:** Ok, we added the word "main".

p. 1, l. 29–p. 2, l. 16: this whole paragraph convey interesting information but is not enough focused for the present study. It can be shortened and simplified by synthesizing the main ideas.

**Response:** Agreed, we will simplify the text.

p. 2, l. 3–4: useless information. In the context of vegetation growth, available N mostly derives from organic matter decomposition (unless the plant is a N-fixer), and available P both from weathering and organic matter decomposition.

**Response:** Agreed and deleted.

p. 2, l. 4–6: not necessary in the context of this article.

**Response:** Agreed and deleted.

p. 2, l. 8–9: N–P interaction is a bit cryptic (is that a statistical term?), can you say something more functional? I think the idea is that the coupling between the N and P cycles drives nutrient limitation.

**Response:** Agreed. We changed the sentence to:

*"The ratio of soil N and P is significant for forest growth on a global scale."*

p. 2, l. 11: move the comma: "In boreal N-limited forests, ..."

**Response:** Ok, done.

p. 2, l. 18: replace "soil nutrients" by "soil nutrient content" or "soil total N and P".

**Response:** Ok, changed to "soil total N and P".

p. 2, l. 27–31: this is not related to your study and could be considered as a confounding factor hindering potential interesting relations. Move the information to the material and methods, and state clearly that you assume reindeer pressure (grazing, trampling, but also nutrient exports or inputs) is not such a confounding factor for this study. Of course, it has to be the case! Did you evaluate somehow the reindeer pressure at your study sites? How? Was it important? Was it typical for the region? Was it constant across sites?

**Response:** Agreed, we moved the information to the material and methods. We have not evaluated the reindeer pressure anyhow in this study. We now state that:

*"An additional factor affecting vegetation composition at our research site is reindeer herding. Since all our plots were located in areas where reindeer roam freely, we assume that the pressure caused by grazing and trampling is equal at all plots."*

p. 2, l. 32: what do you mean by "undisturbed"? The current (steady) state? A baseline status?

**Response:** We changed the word undisturbed to baseline status.

End of the introduction: rephrase and clarify objective(s).
p. 3, l. 1: move this (= method) after the effects of mining.

**Response:** Done.

p. 3, l. 3–4: too much! Focus on mining and keep reindeer and/or climate change for an opening in the discussion. Other option: start the last paragraph of the intro saying that several disturbances such as

mining, grazing (is there a change in grazing in the region? why?), and climate change could affect nutrient status of the ecosystem and you aim at establishing a baseline to monitor the effects of those disturbances. In that case, keep something on reindeer and/or add something on climate change, but keep it short and focused (effects on nutrient status of ecosystem).

p. 3, l. 4–5: simplify and shorten! It is not very clear why you did this... Did you want to establish a relatively simple and cheap protocol of monitoring? For example, surveys of key understory species abundance that would be indicative of the ecosystem nutrient status?

**Response:** As a common response to the previous comments. We have reformulated the last paragraph of the introduction section based on these comments. The exact wording may still change when we further edit the text, but the idea of the paragraph is now this:

*"The general aim of this study was to investigate the baseline status of the forest ecosystem in the Sokli area, for the possible situation there is a need to monitor the effects of phosphate mining. Phosphate mining can cause, for instance, aerial deposition of heavy metals and phosphate to the surroundings of the mine (Reta et al. 2018). Vegetation, soil and foliage chemistry surveys are relatively easy and provide data on the current state of the ecosystem (from the year 2015) that can be used as a reference level for the changes. As the region itself is remote and difficult to access, the methods need to remain simple. The aerial deposition from the mine might increase the contents of P and heavy metals in the top layers of soil, such as the humus layer. This could change the abundance and species composition of the understory vegetation, as the roots of these species are mainly in the humus layer. We aim to explain the understory species composition and abundance with environmental variables, and particularly soil total P content in the humus layer. In addition, we want to figure out which environmental variables could explain soil nutrient contents, especially total P content."*

p. 3, l. 6–7: this sentence does not justify these hypotheses. Delete and replace by something like "We hypothesized: a)... b)...", and move that at the end of the preceding paragraph.

**Response:** Agreed and changed to what was suggested by the referee.

p. 3, l. 9–11: are these hypotheses justified with the preceding text of the introduction? Do they really relate to the objective? Why to focus suddenly on "humus"? I feel some pieces of the reasoning are missing.

p. 3, l. 11: wouldn't it be rather the humus layer that reflects the nutrient content of the leaves? Unless you assume most of the tree uptake occurs in this layer.

**Response:** We agree with these two comments above. We have reformed the hypotheses to include the following information (the exact wording may still change when further editing the text)

*"We hypothesized, that:*
*a)        N and P contents of the soil humus layer correlate with the abundance and species composition of the understory vegetation*
*b)        The topmost soil layers reflect the N and P contents of needle and leaf biomass*
*c)        The topmost soil layer N and P contents reflect/are related to the dominant tree species of the research plots"*

p. 3, l. 14: what are these transects? Are they organized along a gradient? Which one? It seems on the map that it would be the distance to the carbonatite massif.

**Response:** We have now changed this (see previous reply).

p. 3, l. 15: "No plots were located inside the mining district", why?

**Response:** Accessing and doing research at the mining district would have required a permit from the mining company. We found it easier to have our plots on the surrounding land, which is owned by the state of Finland. We now shortly mention this in the materials and methods in the following way:

*"No plots were located inside the mining district, as accessing and doing research at the mining district would have required a permit from the mining company."*

p. 3, l. 19: needs reference, but isn't that what you want to study? Consider moving this info to the discussion.

**Response:** Agreed, we corrected the place of the reference, as it was by mistake in the end of the following sentence. We also moved this info to discussion.

p. 3, l. 22: start the sentence with "Thus,"

**Response:** Done.

p. 3, l. 23–24: delete ", but they were not on any Natura area".

**Response:** Done.

p. 3, l. 26: what is the gradient?

**Response:** The gradient of how far from the mining district all the dust and dirt go. Currently there is no gradient, but in the future there might be.

p. 3, l. 32: add a comma after "5∘C".

**Response:** Ok, done.

p. 4, l. 1: change subsection title to "Plot setup and vegetation characterization".

**Response:** Agreed and changed.

p. 4, l. 5: change subsection title to "Sampling of soil".

**Response:** Agreed and changed.

p. 4, l. 6–10: move this to the preceding section.

**Response:** Agreed and moved.

p. 4, l. 6: cite Table 3 and add to the table tree height and diameter info.

**Response:** Agreed and done. We redid Table 3 so that it includes the following information in their own columns: plot, trees/ha, basal area, total volume of trees, volume of pine, volume of spruce, volume of birch, mean dbh of pine, mean dbh of spruce and mean dbh of birch.

p. 4, l. 7: precise "cover (% surface area)".

**Response:** Agreed and done.

p. 4, l. 8: cite appendix C.

**Response:** Agreed and done.

p. 4, l. 10: add comma after "Altogether".

**Response:** Done.

p. 4, l. 11: provide diameter of the soil corer.

**Response:** Added here that the dimeter of the corer is 5 cm.

p. 4, l. 11: change to "The soil was sampled within one meter from the subplots".

**Response:** Agreed and changed.

p. 4, l. 12: change "The samples" to "The soil cores".

**Response:** Agreed and changed.

p. 4, l. 14–15: simplify and shorten.

**Response:** Agreed. The text now says:

*"The rocky soil and shallow humus layer made it impossible to sample the mineral soil layers in some clusters."*

p. 4, l. 16: remove "already".

**Response:** Done.

p. 4, l. 18: remove "samples from".

**Response:** Ok, done.

p. 4, l. 20–21: two first sentences useless, delete and add "2015" after "September" in the third sentence.

**Response:** Agreed and changed.

p. 4, l. 30: remove ", totalling of 100 leaves per plot".

**Response:** Ok, done.

p. 5, l. 2: replace "in a similar way than needles" by "at 65∘C for 48 h".

**Response:** Ok, done.

p. 5, l. 2: concretely, how did you clean the leaves? With a brush? Deionized water? Other?

**Response:** We have now added the following information here:

*"The needles and few soil particles, which were attached on the litter leaves, were removed with tweezers. The green leaves did not need cleaning, as they were taken directly from the tree. The litter leaves were also rather clean, as it had rained at the time of sampling"*

p. 5, l. 11: fix: "two to three mg of sample were..."

**Response:** Ok, done

p. 5, l. 11: "VarioMax analyser" is a machine, what is the method behind?

**Response:** We have changed this so that it says:

*"…with an element analyzer, which uses high temperature combustion method with subsequent gas analysis of CN (VarioMax)."*

p. 5, l. 12: idem, replace "MilliQ water" by "ultrapure water".

**Response:** Ok, changed.

p. 5, l. 21: "Ordination pattern of the study plots and weighted averages of plant species", not clear, I thought you ordinated the weighted averages.

**Response:** Yes, we ordinated the weighted averages of plant species and based on that the study plots were arranged in a certain way in the ordination space. This has now been clarified to the text.

p. 5, l. 23: state why you did not present results for dim 2 vs dim 3.

**Response:** The text now says:

*"We analysed the data in three-dimensional space, but present the results in 1 vs. 2 and 1 vs. 3 dimensions as the results in 2 vs. 3 dimensions did not give us any new information."*

p. 5, l. 24: "some other environmental variables", list them.

**Response:** We replaced "some other environmental variables" with the environmental variables that we used in the analyses.

p. 6, l. 5: "... imply rather high variation between and within plots", isn't that what you want to study? Maybe a deeper analysis, including tree species, forest stand age, and/or geology could help explaining a bit the variability observed.

**Response:** Yes it is. And now we have more analyses as well.

p. 6, l. 5–6: "Other soil elements...", precise which ones or delete the sentence.

**Response:** Agreed and deleted.

p. 6, l. 11–12: the higher P content in young needles may indicate reallocation processes, which could be discussed shortly in regard to P availability for example.

**Response:** Agreed and done

p. 6, l. 12: replace "Unlike the expectations" by "Unlike our expectations".

**Response:** Agreed and done.

p. 6, l. 15–23: refer to the Table or Fig. these informations are presented (if you go back to Table 2 after referring to Table B2, for example).

**Response:** Thanks for pointing this out, we will do the final check with this just before re-submitting the manuscript.

p. 6, l. 19: replace "discovered" by "detected".

**Response:** Ok, done.

p. 6, l. 22: "Number of species", is this a good variable for your objectives? Could the total % cover of different species groups (the ones of Fig. 5, for example) be more informative?

**Response:** Number of species was originally chosen so that it would be easy to follow if new species start growing or if the variety of species declines in the plots. Sometimes the total % cover might very small, if there is for example only a couple of plants belonging to the group of grasses and sedges in the plot. The % cover was estimated visually and human eyes can make mistakes, especially if the % cover is small and the possible changes are also small. However, if one species starts spreading so that its % cover increases, counting the number of species cannot take this into account. We have now added the total % cover (same groups than for the number of species) and redone the ordination and the figure 7d.

p. 6, l. 27: how do you define the left and right sides of the plots? Is there a threshold or is this empirical?

**Response:** We clarified this in the text in the following way:

*"Plots positioned more on the left-hand side in the figure had higher number of forbs and grasses growing on them than the plots positioned on the right-hand side in the figure (Fig.6a-b)."*

p. 6, l. 28: do you mean Fig. 6b? Or 6a–b?

**Response:** 6a-b, this is now correcred

p. 6, l. 30: no need to cite Table 3 here.

**Response:** Ok, deleted.

p. 7, l. 10: Start the paragraph by citing the Fig.: "In Fig. 7c–d, ..."

**Response:** Agreed and done.

p. 7, l. 18: did you analyze soil samples by plot or by cluster? Did you quantify both within- and between-plot variabilities?

**Response:** We analyzed the soil samples by cluster, but have used averages of the whole plot and quantified the between-plot variabilities only. This sentence is now corrected.

p. 7, l. 19: refer to Fig. 3 where the soil P contents are presented. Also, this is a huge variability: is mainly due to between- or within-plot variation? If this is between plots, you might be able to explain it somehow by additional exploration of environmental factors, but if it is within plot there is no hope tree species will help, for example...

**Response:** This variability is due to between-plot variation. As mentioned in earlier responses, tree species is now also included.

p. 7, l. 24: "implying that decaying plant parts were a major source of P", for what? The soil organic layers or plants?

**Response:** As addition of P to the soil organic layer. This has now been clarified.

p. 7, l. 24–27: would it be possible to find a pattern of P content in the deep soil layers according to the soil rock parent material/geology? Do you think that high P content in the humus layer is important for plant nutrition (recycling) or is that just high litter production coupled with slow decomposition rates?

**Response:** We included soil parent material to the linear mixed effect models, but it seemed to have very little importance compared to the other fixed effects.  According to our results, high P content in the humus is related with increased coverage of grasses and sedges, which means it is important for plant nutrition. It is also a result of high litter production (from birch) and slow decomposition rates.

p. 7, l. 28: what is the context of the study by Köster et al (2014)?

**Response:** They conducted their study in the same region, although not at the same plots. We have now added a sentence about this.

p. 8, l. 2: and so? Can you say something concrete for your study area?

**Response:** We ended up deleting this from the discussion, as it did not seem relevant anymore.

p. 8, l. 4: replace "similar than" by "similar to".

**Response:** Ok, done.

p. 8, l. 18–19: Which analysis/which Fig. or Table? Table 4? But is it species richness or ordination pattern which was regressed? What do you mean exactly by "species richness"? The number of species?

**Response:** We added reference to Table 4, as well as Fig. 6b &d. We are talking about the % cover and number of species in the group of grasses and sedges. We have now replaced the word richness with the previous explanation.

p. 8, l. 19–20 and l. 30: which soil layer(s) are you considering? The hypotheses were about humus.

**Response:** This was about humus, we have now changed these.

p. 8, l. 32–p. 9, l. 2: "needles were sampled at different time of year than soil...", this should be mentioned in, and even might only be part of, the materials and method section. If you sampled the needle at a right moment, it should be quite integrative of nutrient availability across the growing season.

**Response:** We have moved this information to the materials and methods section, as suggested.

p. 9, l. 4–8: why not such an opening but right now it is not well connected to what precedes. You could also talk about the coupling between N and P cycles and how this could be affected by climate change or disturbances and in turn affect ecosystem status and processes.

**Response:** Agreed, we have changed the last paragraph of the discussion so that it is more connected to what is discussed in the earlier paragraphs. We added discussion "about the coupling between N and P cycles and how this could be affected by climate change or disturbances and in turn affect ecosystem status and processes", as suggested.

p. 9, l. 6: "variation in the vegetation", be more precise (which parameter, sense of variation).

**Response:** Agreed and changed to "change of plant species"

p. 9, l. 10 and 17: "vegetation dynamics", this is not what you study here, change to "vegetation community composition" or something like that.

**Response:** Agreed and changed.

p. 9, l. 12: change "has been discovered" to "was found".

**Response:** Agreed and changed.

Fig. 2: draw or remind in the title where tree cover was described.

**Response:** We have now added to the caption that trees were measured from the whole 30 x 30 m area that the clusters delineate. We also drew marks to the figure to make it clearer where the outer borders of the 30 x 30 m area were.

Fig. 3: would it be possible to also represent tree cover species (or different groups based on dominant species)? or age? or geology? Is this graph representing between- plot variability (i.e. you first calculated the mean for each plot and made the boxplots with those means) or a mix between within- and between- plot variability (i.e. you took all sub-plots values to make the boxplot)? It would be interesting to compare between- and within-plot variabilities.

Fig. 4: same comments as for Fig. 3.

**Response:** Figure 3 and 4 represent between-plot variability (i.e. we first calculated the mean for each plot and made the boxplots with those means). We have replaced these figures with box plots, which show the within plot variation in addition to the between plot variation.

Fig. 5: What is the correlation coefficient calculated? (Pearson? Other?) Precise what is the "bottom layer". Why not to call this layer "moss lichen"? It would be clearer.

**Response:** We added that Pearson correlations were used. The bottom layer includes mosses and lichen, so we re-named this layer "moss lichen" and redid the figure.

Fig. 6: what is the criteria to define "the most abundant species"?

**Response:** We added here that most abundant species here means those species, which have the highest % coverage.

Fig. 6, l. 8: replace "generic" by "genera". Start the last sentence of the figure legend by "In (a),"

**Response:** Agreed and done.

Fig. 7: remind which soil layer was considered for this analysis (I am assuming it's O).

**Response:** Yes, it is O layer and we added this information to the figure caption.

Table 1: "degree days", shouldn't that be called "sum of degree days"? What is the unit?

**Response:**  Degree days here mean the growing degree days, whose unit is GDD or °C days.  We changed it to "growing degree day sum" as well added how it is calculated:

*"Growing degree day sum was calculated as the mean daily temperature (average of daily maximum and minimum temperatures) above 5 °C base temperature, accumulated on a daily basis over the year. Negative values are treated as zeros and ignored."*

Table 2: the classical ordering would be C, N, P, K, C:N, N:P. For numbers (mean and sd), provide the same number of digits after the dot for each column (one is enough) and align the numbers to the right to ease comparison of lines. Why not to put K in the same unit as the others?

**Response:** Agreed, we have made these changes.

Table 3: did you estimate the whole aerial volume of trees or just the trunk volume? Make three sub-columns for each species abundance. Add in this table tree height, diameter,...

**Response:** We estimated the trunk volume. We made the suggested changes.

Table 4: remind the first seven lines are soil values.

**Response:** Agreed and done.

Tables A2–A4: From which statistical test are these table issued? These tables hardly help to address your objectives.

**Response:** We have deleted these tables and replaced them with a table which includes the fixed effects (from the linear mixed effect models) and their Chisq values, p-values and pseudoR2 values.

Table B1: title: change to "Statistically significant differences between needle age group by species". From which test?

**Response:** We changed the title and added that these are from the one-way ANOVA, with Tukey's HSD for post hoc. We will add this information to the caption.

Appendix C: precise "(% of surface area)".

**Response:** Agreed and done.

---

## Author Response (AR1)

**Response to referee #1**

Below are the comments of the referee #1 in black and our responses in blue font and changes to manuscript *italicized and gray*.

Soil nutrients and stoichiometry is an important topic in forest ecosystems. The manuscript studied the relationships between understory vegetation species abundance in a boreal forest and soil /leaf nutrients. However, there are several major concerns about the statistical methods, the data presented in the figures and tables and shortage of basic information regarding the study site. Additionally, there are also grammatical issues and inappropriate descriptions of the results. The discussion cannot fully support their hypotheses and results.

**Response:** We thank anonymous referee #1 for the time they have spent revising our manuscript. We found the comments very helpful and sincerely appreciate all the detailed and concrete suggestions on how to proceed with the manuscript. As both referees brought up points about the role of tree species in soil nutrient content, we have added analyses and discussion about this topic in our manuscript.

Statistical analysis:
One-way ANOVA were chosen in the manuscript, it implies that the plot was the only factor. However, tree species plays an important role in soil nutrients as well, thus tree species should be considered as a confounding factor. Due to high spatial heterogeneity in soil samples, when you determine the difference in different plots, the block effects also should be considered in the statistical analysis.
Considering the relationships between species composition and soil nutrients, besides the species, the authors also should treat age classes as the second factor. For the same reason, soil layer also should be considered in the analysis.

**Response:** We agree with these comments. Unfortunately, we do not have exact knowledge of the age of the trees at each plot, as we did not core the trees. We know the approximate age of trees at plot A6 and have used this as help, when we estimated the tree ages based on their dbh and put the trees in three different classes (young trees 1-9.9 cm, mid-aged 10-14.9 cm and old > 15 cm). We added information about the tree age classification to materials and methods. We tested the effect of dominant tree species, tree age, soil parent material/bedrock type, and soil horizon on soil P, N and C:N with linear mixed-effect models and have added the following description to section 2.4:

*"We tested the effects of environmental variables on soil P, N and C:N with linear mixed-effect models. We used dominant tree species, estimated tree age, rock parent material and soil horizon as fixed effects and plot as random effect. Soil P needed to be log-transformed while for N and C:N the visual inspection of residual plots did not reveal obvious deviations from homoscedasticity or normality. We obtained p-values for the fixed effects by likelihood ratio tests, where the full model with all the fixed effects was tested against model where each fixed effect was removed in turn. We used package lme 4 (Bates et al. 2015) in R programme 3.4.3 (R Development Core Team, 2017) for building the models. Pseudo R2-value for the models were calculated by using package r2glmm (Jaeger 2017)The models took the form:*

$$SC_{P,N,CN} = B_0 + B_{dt} + B_{ta} + B_g + B_h + \in, \hspace{2cm} (1)$$

*where $SC_{P,N,CN}$ is the soil nutrient content (P, N or C:N ratio), $B_0$ denotes a fixed intercept parameter, $B_{dt}$ denotes the fixed unknown parameters associated with the dominant tree species, $B_{ta}$ denotes the fixed unknown parameters associated with the age of the dominant tree species, $B_g$ denotes the fixed unknown parameters associated with the rock parent material, $B_h$ denotes the fixed unknown parameters associated with soil horizon. The random effect $\in$ is assumed to take the form:*

$$\in = \propto_p + u ,\hspace{5cm} (2)$$

*where $\propto_p$ denotes the random parameters related with the research plot and $u$ is an unobservable error term. Random-effect parameters and random-error term are assumed to follow normal distributions $\propto_p \sim N(0, \sigma_p^2)$ and $u \sim N(0, \sigma_u^2)$."*

We made a new subsection for the results of mixed effect models under section 3 and added a table, which includes the fixed effects and their Chisq values, p-values and pseudoR2 values. To the appendices we added figures of the fitted vs. residuals, q-q plots and histograms of the residuals and removed the unnecessary tables of the previous one-way ANOVAs.

While we agree that the within-plot variation of soil element content is important, it could not be added to this same model, as the other factors were on the plot scale. We made box plots about the within plot variation of soil total P, N and C:N ratio.

In order to more precisely study the relationship of tree species and understory vegetation, we added the volume of birch per plot to the ordination and to Fig. 7d. We also added the cover (% of surface area) of species in the same species groups to the ordination and Fig. 7d.

Shortage of basic information: the authors provided the basic tree and other species composition in the Table 3. The tree age and biomass also affects the soil nutrients. The author should provide the mean basal area, leaf area index and mean DBH. These basic information would be useful to estimate the effects of tree species on the under-story species composition and on soil nutrients.

**Response:** Yes, we agree. We redid this table (now Table 2) so that it includes the following information in their own columns: plot, trees/ha, basal area, total volume of trees, volume of pine, volume of spruce, volume of birch, mean dbh of pine, mean dbh of spruce and mean dbh of birch. Unfortunately, we have no information of LAI.

As to the weather information, the min and max temperature should also be provided in Table 1.

**Response:** Agreed, we added the min and max temperature to the table.

Authors should add a new table/Figure to show the mean soil nutrients in the birch, scots pine and spruce plots in each layer and make stat analysis.

**Response:** We agree. We made such a table and did the statistical analysis related to that. We marked the stat. differences between tree species to the table. We added the following piece of text to section 2.4:

*"We grouped the plots based on their dominant tree species into pine, birch and spruce plots and calculated the average soil nutrient contents in each horizon in these plots. We then compared the nutrient contents in each soil horizon with one-way ANOVA.*

We also added explanation of the results of the ANOVA to section 3.1.

In Fig 5, there were no adj-R2 value to show which factor possessed the most weight. At the same time, these correlations could be better presented in Table not in fig.

**Response:** We corrected the unclear figure caption in Fig 5 to include the following information:

*"Positive correlations are displayed in blue and negative correlations in red. Colour intensity and size of the circle are proportional to the correlation coefficients.."*

The confusing plot numbers in table A2/A3 /A4 and Table B2. In the Table A, the plot number was in alphabetical order while the Arabic number was adopted in table B2.

**Response:** We corrected the confusing and incorrect plot numbering in Table B2.

We cannot find the stats evidence support the data. For example: "Foliar N:P ratio did not show any differences in either species between plots.. . .. . .. . .green leaves compared to other species." (3.2)

**Response:** We changed this piece of text to:

*"On the other hand, N and C contents, as well as the C:N ratio of the conifers showed some between-plot variation ($p < 0.05$), but no significant variation was found in the foliar N:P ratio in either species."*

In the results section, the first sentence in each sub section provides meaningless information for the data and these sentences can be deleted. For example: " The average contents. . .. . .. In fig 4" (3.1 soil element contents). The same was also found in each paragraph.

**Response:** Agreed, we deleted the sentences including meaningless information.

There were some grammatical issues in each paragraph. There was no deep discussion to support the hypotheses and results.

**Response:** We thank the referee for pointing out these issues. A native English speaker has checked our revised manuscript.
We have revised the discussion section based on the comments from both referees. We synthesized and shortened the part of text where we compare our total nutrient contents to previous studies as well as reorganized the sections so that the main results become clear in the first paragraph of the discussion section. We wrote more about the role of tree species in soil P content and highlight how and why our results are important and relate with the previous findings.

**Response to referee #2**

Below are the comments of the referee #2 in black and our responses in blue font and changes to manuscript *italicized and gray*.

We thank anonymous referee #2 for the time they have spent revising our manuscript. We found the comments very helpful and sincerely appreciate all the detailed and concrete suggestions on how to proceed with the manuscript. As both referees brought up points about the role of tree species in soil nutrient content, we added analyses and discussion about this topic in our manuscript.

Title: rephrase; "understory vegetation" is not precise, that are the species composition and abundance that were studied. Why to focus on the understory since the tree cover was also studied? You could also focus on what you consider as your main result, for example "Soil total P explains vegetation community composition in a northern boreal forest ecosystem".

**Response:** We appreciate the suggestion, but feel that the title example by the referee would be too strong of a statement. Our study is more of a case study about a very special area surrounding a phosphate massif and, thus, it would be rather risky to generalize the results to cover also other northern boreal forest ecosystems. Thus, we changed the title to: "Soil total phosphorus and nitrogen explain vegetation community composition in a northern forest ecosystem near a phosphate massif". We feel that it is important to mention the phosphate massif in the title to make it clear that our study area has some special features.

Abstract: needs a sentence on sampling design (the way the relationship were addressed)... In the present case, the reader have no idea what the "plots" refer to and what to conclude from that information. Here and in the materials and methods section, you need to state clearly that you described vegetation, and sampled tree leaves and soil, at different distances from the P ore. Revise this abstract after clarifying the objectives and re-analyzing/discussing the results.

**Response:** Agreed. We have added information about how the study plots and measurements were arranged.

Introduction: could be simplified and shortened. Also needs to better formulate the objective(s) and hypotheses, and/or to provide all the information that lead to such hypotheses. In particular, it is not very clear why the hypotheses focus on the humus layer.

**Response:** Agreed. We have simplified the introduction according to the suggestions and focused on re-formulating the aims and hypotheses.

By comparing the title, abstract and introduction, it is not clear if the objective if finally (1) to explain the understory species composition and abundance with environmental parameters, and particularly soil total P content, or (2) to predict soil/environment nutrient status by surveying understory vegetation. One option should be chosen and the whole article built around.

**Response:** This is a very good point and helped us to clarify the "common thread" of the manuscript. We originally started with option 1) and it is still valid. We aim to explain the understory species composition and abundance with environmental parameters, and see if soil total P and N content have an effect on them. In addition, we want to figure out what environmental parameters could explain soil N and P contents.

Material and methods: The site selection process is not clear; in particular, what is the basis for selecting those transects? I did not get if there is any gradient, for example.

**Response:** We have added the following sentences to materials and methods (after "We established 16 study plots…):

*" The plots were located different distances from the phosphate ore in four transects, enabling evaluation of the possible effects of the mine in the future."*

Are all the study sites located at a similar elevation with similar climate conditions?

**Response:** Yes they are.

As the study sites were located on different geological units (Fig. 1), did the authors tried to include such factor in their analysis?

**Response:** We have now included the geological unit to the analyses.

Do we know anything about the P contents of those rocks? Are these rocks essential parent material for the soils developed at the sites? Any idea of the age/development stage of the soils?

**Response:** The Sokli phosphate ore has been carefully sampled and studied, but the surrounding area lacks such detailed information. We added to the figure caption of Fig 1. that this map shows the bedrock in the region, which is essential parent material for the soil. The bedrock in Finland is among the oldest in Europe but the soils have been modified by the latest ice age. Sokli is different from the surrounding soils also because it is located in a sheltered depression and has not been affected by the erosion caused by latest ice age.

Are all the soils studied haplic podzols?

**Response:** Yes they are.

As for the statistical analyses, it seems the forest stand composition could be better taken into account by accounting for the species % of volume or by grouping sites according to their dominant species. As raised by reviewer 1, stand age could also be a confounding factor.
I think the authors could try to better explain the variations they observe in soil values. Also, why did the authors focused in the elemental contents of the O horizon (humus) in their analysis? Do we have an idea of the distribution of fine roots (and vegetation uptake zone) in the soil profile?

**Response:** We agree with these comments. We originally chose humus as the roots of understory vegetation are mainly in this layer, but we have now done more stat. analyses including other layers as well.
Unfortunately, we do not have exact knowledge of the age of the trees at each plot, as we did not core the trees. We know the approximate age of trees at plot A6 and have used this as help, when we estimated the tree ages based on their dbh and put the trees in three different classes (young trees 1-9.9 cm, mid-aged 10-14.9 cm and old > 15 cm). We added information about the tree age classification to section 2.2.1. We tested the effect of dominant tree species, tree age, soil parent material/bedrock type, and soil horizon on soil P, N and C:N with linear mixed-effect models and have added the following description to section 2.4:

*"We tested the effects of environmental variables on soil total P and N contents and C:N with linear mixed-effect models. We used dominant tree species, estimated age class, rock parent material (Fig. 1) and soil*

*horizon as fixed effects and plot as random effect. Soil total P needed to be log-transformed, while for N and C:N the visual inspection of residual plots did not reveal obvious deviations from homoscedasticity or normality. We obtained p-values for the fixed effects by likelihood ratio tests, where the full model with all the fixed effects was tested against a model where each fixed effect was removed in turn. We used package lme4 (Bates et al. 2015) in R programme 3.4.3 (R Development Core Team 2017) for building the models. Pseudo R2-values for the models were calculated by using package r2glmm (Jaeger 2017). The models took the form:*

$$SC_{P,N,CN} = B_0 + B_{dt} + B_{ta} + B_g + B_h + \in, \qquad (1)$$

*where $SC_{P,N,CN}$ is the soil nutrient content (P, N or C:N ratio), $B_0$ denotes a fixed intercept parameter, $B_{dt}$ denotes the fixed unknown parameters associated with the dominant tree species, $B_{ta}$ denotes the fixed unknown parameters associated with the age of the dominant tree species, $B_g$ denotes the fixed unknown parameters associated with the rock parent material, $B_h$ denotes the fixed unknown parameters associated with the soil horizon. The random effect $\in$ is assumed to take the form:*

$$\in = \propto_p + u, \qquad (2)$$

*where $\propto_p$ denotes the random parameters related with the research plot and u is an unobservable error term. Random-effect parameters and random-error term are assumed to follow normal distributions $\propto_p \sim N(0, \sigma_p^2)$ and $u \sim N(0, \sigma_u^2)$."*

We made a new subsection for the results of mixed effect models under section 3 and added a table, which includes the fixed effects and their Chisq values, p-values and pseudoR2 values. To the appendices we added figures of the fitted vs. residuals, q-q plots and histograms of the residuals and removed the unnecessary tables of the previous one-way ANOVAs.

In order to more precisely study the relationship of tree species and understory vegetation, we added the volume of birch per plot to the ordination and to Fig. 7d. We also added the cover (% of surface area) of species in the same species groups to the ordination and Fig. 7d.

Additionally (based on a suggestion from referee #1), we grouped the plots based on their dominant tree species and calculated the means of soil nutrients and ratios in each soil horizon in pine, birch and spruce plots. We made a table of these and compared the nutrient contents in each soil horizon with one-way ANOVA.

The use of understory species is interesting, would it be possible to go further by narrowing down the number of species, by detecting indicator species (of the P status for example), and building a "simple" prediction model?

**Response:** Indeed, this would be interesting. However, we feel that building such a model would require a lot more study plots and data.

Results: I feel some results are not presented in the way that best help to address the questions of interest. I think in particular about the Fig. 3 and 4 or Table A2–A4, where we don't have any clue about what could lead the variability and differences (forest stand composition? rock parent material? other?).

**Response:** Agreed. We have replaced Fig. 3 and 4. with box plots which show the within plot variation in these soil nutrients. We have also deleted Tables A2-A4, as the new table of fixed effects and figures of residuals and q-q plots (described above) are more useful in this context.

Discussion: The authors have put honorable efforts in comparing the data they obtained with known ranges of values for similar areas published in the literature. However, this part of the discussion could be better synthesized and written in a simpler and shorter style. The discussion lacks development on the results obtained in regard to the objectives of the study. What is the functional significance of these results? When focusing on key results, the Fig. and/or Tables where they are presented should be reminded to help the reader.

**Response:** We thank the referee for this comment.  We have revised the discussion section based on the comments from both referees.  We synthesized and shortened the part of text where we compare our total nutrient contents to previous studies as well as reorganized the sections so that the main results become clear in the first paragraph of the discussion section. We wrote more about the role of tree species in soil P content and highlight how and why our results are important and relate with the previous findings. We have also paid more attention on referring to the Figs and Tables.

Conclusions: potentially revise according to modifications in the introduction and discussion.

**Response:** We have modified the conclusions.

Technical corrections
Whole text: refer to Fig., Tables, and Appendix when appropriated, and order and number them following their first apparition in the text.

**Response:** Done.

p. 1, l. 8: "We studied the relationship of forest understory vegetation with nutrient contents of soil and tree leaves...": write something which fits with your objectives and title. Again, "understory vegetation"is not precise, add "species composition and abundance", which appears only at l. 12–13.

p. 1, l. 9–10: add a comma: "At most study plots, boreal..."

p. 1, l. 11–12: here and elsewhere, change "abundance and species composition of the vegetation" to "species composition and abundance of the understory vegetation".

p. 1, l. 13–14: what is the information you want to raise?

p. 1, l. 19: some fixes: "... controlling the species compositions of tree stand and understory..."

**Response:** The previous comments were related to the abstract, which has now been revised.

p. 1, l. 21: what do you mean by "modified"? I would rather say that those ecosystems are "characteristically cold, have a short growing season, and are nutrient-poor".

**Response:** Agreed and changed.

p. 1, l. 21–22: "affects" is not precise, and the sentence could be shorter and simpler. Suggestion: "Organic matter decomposition and nutrient release are usually slow in cold climates".

**Response:** Agreed and changed

p. 1, l. 22–24: not very informative.

**Response:** Agreed, we deleted this sentence.

p. 1, l. 25: change "tree species affect" by "tree cover affects", unless you want to precise "different tree species affect differently understory..." (if this fits to the references cited).

**Response:** Agreed, and changed to tree cover.

p. 1, l. 26: again, "understory vegetation" is not precise, which parameters? Focus on what you study here (i.e. species composition and abundance).

**Response:** Agreed, we changed "understory vegetation" to "species composition and abundance in the understory"

p. 1, l. 27: "litterfall" is sufficient since it also includes branches, etc.

**Response:** Agreed and deleted the word "leaf".

p. 1, l. 29: N and P "are generally the main growth-limiting nutrients..."

**Response:** Ok, we added the word "main".

p. 1, l. 29–p. 2, l. 16: this whole paragraph convey interesting information but is not enough focused for the present study. It can be shortened and simplified by synthesizing the main ideas.

**Response:** Agreed, we have revised the text.

p. 2, l. 3–4: useless information. In the context of vegetation growth, available N mostly derives from organic matter decomposition (unless the plant is a N-fixer), and available P both from weathering and organic matter decomposition.

**Response:** Agreed and deleted.

p. 2, l. 4–6: not necessary in the context of this article.

**Response:** Agreed and deleted.

p. 2, l. 8–9: N–P interaction is a bit cryptic (is that a statistical term?), can you say something more functional? I think the idea is that the coupling between the N and P cycles drives nutrient limitation.

**Response:** Agreed. We changed the sentence to:

*"The ratio of soil N to soil P is significant for forest growth on a global scale."*

p. 2, l. 11: move the comma: "In boreal N-limited forests, ..."

**Response:** Ok, done.

p. 2, l. 18: replace "soil nutrients" by "soil nutrient content" or "soil total N and P".

**Response:** We have revised this part of text.

p. 2, l. 27–31: this is not related to your study and could be considered as a confounding factor hindering potential interesting relations. Move the information to the material and methods, and state clearly that you assume reindeer pressure (grazing, trampling, but also nutrient exports or inputs) is not such a confounding factor for this study. Of course, it has to be the case! Did you evaluate somehow the reindeer pressure at your study sites? How? Was it important? Was it typical for the region? Was it constant across sites?

**Response:** Agreed, we moved the information to the material and methods. We have not evaluated the reindeer pressure anyhow in this study.

p. 2, l. 32: what do you mean by "undisturbed"? The current (steady) state? A baseline status?

**Response:** We added baseline status after the word undisturbed.

End of the introduction: rephrase and clarify objective(s).
p. 3, l. 1: move this (= method) after the effects of mining.

**Response:** Done.

p. 3, l. 3–4: too much! Focus on mining and keep reindeer and/or climate change for an opening in the discussion. Other option: start the last paragraph of the intro saying that several disturbances such as mining, grazing (is there a change in grazing in the region? why?), and climate change could affect nutrient status of the ecosystem and you aim at establishing a baseline to monitor the effects of those disturbances. In that case, keep something on reindeer and/or add something on climate change, but keep it short and focused (effects on nutrient status of ecosystem).

p. 3, l. 4–5: simplify and shorten! It is not very clear why you did this... Did you want to establish a relatively simple and cheap protocol of monitoring? For example, surveys of key understory species abundance that would be indicative of the ecosystem nutrient status?

**Response:** As a common response to the previous comments. We have reformulated the last paragraph of the introduction section based on these comments, and it now says:

*"The general aim of this study was to determine the undisturbed baseline status of the forest ecosystem in terms of soil, understorey vegetation and tree layers in the Sokli area in case there is a need to monitor the effects of phosphate mining. Phosphate mining can cause, for instance, aerial deposition of heavy metals and phosphate onto the surroundings of the mine (Reta et al. 2018), which can lead to changes in the*

*abundance and species composition of the understorey. Vegetation, soil and foliage chemistry surveys provide data on the current state of the ecosystem (from the year 2015) that can be used as a reference level for the changes. Our specific aim was to identify which factors in the soil and tree layer explain the composition and abundance of plant species. In addition, we studied which environmental variables could explain soil nutrient contents, especially total P content."*

p. 3, l. 6–7: this sentence does not justify these hypotheses. Delete and replace by something like "We hypothesized: a)... b)...", and move that at the end of the preceding paragraph.

p. 3, l. 9–11: are these hypotheses justified with the preceding text of the introduction? Do they really relate to the objective? Why to focus suddenly on "humus"? I feel some pieces of the reasoning are missing.

p. 3, l. 11: wouldn't it be rather the humus layer that reflects the nutrient content of the leaves? Unless you assume most of the tree uptake occurs in this layer.

**Response:** We agree with the comments above and have reformed the hypotheses in the following way:

*"We hypothesize that there are positive relationships between:*
*a)        N and P contents of the soil humus layer and the abundance and species composition of the understory vegetation*
*b)        N and P contents in the topmost soil layers and the N and P contents of needle and leaf biomass*
*c)        N and P contents in the topmost soil layers and the occurrence of birch trees in the research plots"*

p. 3, l. 14: what are these transects? Are they organized along a gradient? Which one? It seems on the map that it would be the distance to the carbonatite massif.

**Response:** We have changed this so that it says:

*"The plots were located different distances from the phosphate ore in four transects, enabling evaluation of the possible effects of the mine in the future."*

p. 3, l. 15: "No plots were located inside the mining district", why?

**Response:** Accessing and doing research at the mining district would have required a permit from the mining company. We found it easier to have our plots on the surrounding land, which is owned by the state of Finland. We now shortly mention this in the materials and methods in the following way:

*"No plots were located inside the mining district, as accessing and doing research at the mining district would have required a permit from the mining company."*

p. 3, l. 19: needs reference, but isn't that what you want to study? Consider moving this info to the discussion.

**Response:** Agreed, we corrected the place of the reference, as it was by mistake in the end of the following sentence. We also moved this info to discussion.

p. 3, l. 22: start the sentence with "Thus,"

**Response:** Done.

p. 3, l. 23–24: delete ", but they were not on any Natura area".

**Response:** Done.

p. 3, l. 26: what is the gradient?

**Response:** The gradient of how far from the mining district all the dust and dirt go. Currently there is no gradient, but in the future there might be.

p. 3, l. 32: add a comma after "5∘C".

**Response:** Ok, done.

p. 4, l. 1: change subsection title to "Plot setup and vegetation characterization".

**Response:** Agreed and changed.

p. 4, l. 5: change subsection title to "Sampling of soil".

**Response:** Agreed and changed.

p. 4, l. 6–10: move this to the preceding section.

**Response:** Agreed and moved.

p. 4, l. 6: cite Table 3 and add to the table tree height and diameter info.

**Response:** Agreed and done. We redid this table (now Table 2) so that it includes the following information in their own columns: plot, trees/ha, basal area, total volume of trees, volume of pine, volume of spruce, volume of birch, mean dbh of pine, mean dbh of spruce and mean dbh of birch.

p. 4, l. 7: precise "cover (% surface area)".

**Response:** Agreed and done.

p. 4, l. 8: cite appendix C.

**Response:** Agreed and done (now appendix A).

p. 4, l. 10: add comma after "Altogether".

**Response:** Done.

p. 4, l. 11: provide diameter of the soil corer.

**Response:** Added here that the dimeter of the corer is 5 cm.

p. 4, l. 11: change to "The soil was sampled within one meter from the subplots".

**Response:** We changed this to:

*" The soil was sampled within a 1m distance from the subplots"*

p. 4, l. 12: change "The samples" to "The soil cores".

**Response:** Agreed and changed.

p. 4, l. 14–15: simplify and shorten.

**Response:** Agreed. The text now says:

*"The rocky soil and shallow humus layer made it impossible to sample the mineral soil layers in some clusters."*

p. 4, l. 16: remove "already".

**Response:** Done.

p. 4, l. 18: remove "samples from".

**Response:** Ok, done.

p. 4, l. 20–21: two first sentences useless, delete and add "2015" after "September" in the third sentence.

**Response:** Agreed and changed.

p. 4, l. 30: remove ", totalling of 100 leaves per plot".

**Response:** Ok, done.

p. 5, l. 2: replace "in a similar way than needles" by "at 65∘C for 48 h".

**Response:** Ok, done.

p. 5, l. 2: concretely, how did you clean the leaves? With a brush? Deionized water? Other?

**Response:** We have now added the following information here:

*"The needles and the few soil particles attached on the litter leaves, were removed with tweezers. The green leaves did not need cleaning. The litter leaves were also rather clean, as it had rained at the time of sampling"*

p. 5, l. 11: fix: "two to three mg of sample were..."

**Response:** We changed this to:

*"Samples of 2–3 mg were measured and analysed…"*

p. 5, l. 11: "VarioMax analyser" is a machine, what is the method behind?

**Response:** We have changed this so that it says:

*"…with an element analyzer, which uses high temperature combustion method with subsequent gas analysis of CN (VarioMax, Elementar Analysensysteme GmbH, Germany)."*

p. 5, l. 12: idem, replace "MilliQ water" by "ultrapure water".

**Response:** Ok, changed.

p. 5, l. 21: "Ordination pattern of the study plots and weighted averages of plant species", not clear, I thought you ordinated the weighted averages.

**Response:** We changed this to:

*" Ordination pattern of the plots based on the Bray-Curtis dissimilarity indices in floristic composition was analysed to find the main environmental gradients behind the vegetation variation. "*

p. 5, l. 23: state why you did not present results for dim 2 vs dim 3.

**Response:** The text now says:

*"We analysed the data in three-dimensional space but present the results in 1 vs 2 and 1 vs 3 dimensions (the results in 2 vs 3 dimensions did not give any new information)."*

p. 5, l. 24: "some other environmental variables", list them.

**Response:** We replaced "some other environmental variables" with the environmental variables that we used in the analyses.

p. 6, l. 5: "... imply rather high variation between and within plots", isn't that what you want to study? Maybe a deeper analysis, including tree species, forest stand age, and/or geology could help explaining a bit the variability observed.

**Response:** Yes it is. And now we have more analyses as well.

p. 6, l. 5–6: "Other soil elements...", precise which ones or delete the sentence.

**Response:** Agreed and deleted.

p. 6, l. 11–12: the higher P content in young needles may indicate reallocation processes, which could be discussed shortly in regard to P availability for example.

**Response:** Agreed and done

p. 6, l. 12: replace "Unlike the expectations" by "Unlike our expectations".

**Response:** We changed this to "against our expectations".

p. 6, l. 15–23: refer to the Table or Fig. these informations are presented (if you go back to Table 2 after referring to Table B2, for example).

**Response:** Thanks for pointing this out!

p. 6, l. 19: replace "discovered" by "detected".

**Response:** Ok, done.

p. 6, l. 22: "Number of species", is this a good variable for your objectives? Could the total % cover of different species groups (the ones of Fig. 5, for example) be more informative?

**Response:** Number of species was originally chosen so that it would be easy to follow if new species start growing or if the variety of species declines in the plots. Sometimes the total % cover might very small, if there is for example only a couple of plants belonging to the group of grasses and sedges in the plot. The % cover was estimated visually and human eyes can make mistakes, especially if the % cover is small and the possible changes are also small. However, if one species starts spreading so that its % cover increases, counting the number of species cannot take this into account. We have now added the total % cover (same groups than for the number of species) and redone the ordination and the figure 7d.

p. 6, l. 27: how do you define the left and right sides of the plots? Is there a threshold or is this empirical?

**Response:** We clarified this in the text in the following way:

*"Plots positioned more on the left-hand side in the figure had a higher number of forbs and grasses growing on them than the plots positioned on the right-hand side in the figure (Fig.6a,b)."*

p. 6, l. 28: do you mean Fig. 6b? Or 6a–b?

**Response:** 6a-b, this is now correcred

p. 6, l. 30: no need to cite Table 3 here.

**Response:** Ok, deleted.

p. 7, l. 10: Start the paragraph by citing the Fig.: "In Fig. 7c–d, ..."

**Response:** We have changed the first sentence of the paragraph.

p. 7, l. 18: did you analyze soil samples by plot or by cluster? Did you quantify both within- and between-plot variabilities?

**Response:** We analyzed the soil samples by cluster, but have used averages of the whole plot and quantified the between-plot variabilities only. This sentence is now corrected.

p. 7, l. 19: refer to Fig. 3 where the soil P contents are presented. Also, this is a huge variability: is mainly due to between- or within-plot variation? If this is between plots, you might be able to explain it somehow by additional exploration of environmental factors, but if it is within plot there is no hope tree species will help, for example...

**Response:** This variability is due to between-plot variation. As mentioned in earlier responses, tree species is now also included.

p. 7, l. 24: "implying that decaying plant parts were a major source of P", for what? The soil organic layers or plants?

**Response:** As an addition of P to the soil organic layer. This has now been clarified.

p. 7, l. 24–27: would it be possible to find a pattern of P content in the deep soil layers according to the soil rock parent material/geology? Do you think that high P content in the humus layer is important for plant nutrition (recycling) or is that just high litter production coupled with slow decomposition rates?

**Response:** We included soil parent material to the linear mixed effect models, but it seemed to have very little importance compared to the other fixed effects. According to our results, high P content in the humus is related with increased coverage of grasses and sedges, which means it is important for plant nutrition. It is also a result of high litter production (from birch) and slow decomposition rates.

p. 7, l. 28: what is the context of the study by Köster et al (2014)?

**Response:** They conducted their study in the same region, although not at the same plots. We have now added a sentence about this.

p. 8, l. 2: and so? Can you say something concrete for your study area?

**Response:** We ended up deleting this from the discussion, as it did not seem relevant anymore.

p. 8, l. 4: replace "similar than" by "similar to".

**Response:** Ok, done.

p. 8, l. 18–19: Which analysis/which Fig. or Table? Table 4? But is it species richness or ordination pattern which was regressed? What do you mean exactly by "species richness"? The number of species?

**Response:** We are talking about the % cover and number of species in the group of grasses and sedges. We have now replaced the word richness with the previous explanation and otherwise revised this part of text.

p. 8, l. 19–20 and l. 30: which soil layer(s) are you considering? The hypotheses were about humus.

**Response:** This was about humus, we have now changed these.

p. 8, l. 32–p. 9, l. 2: "needles were sampled at different time of year than soil...", this should be mentioned in, and even might only be part of, the materials and method section. If you sampled the needle at a right moment, it should be quite integrative of nutrient availability across the growing season.

**Response:** We have moved this information to the materials and methods section, as suggested.

p. 9, l. 4–8: why not such an opening but right now it is not well connected to what precedes. You could also talk about the coupling between N and P cycles and how this could be affected by climate change or disturbances and in turn affect ecosystem status and processes.

**Response:** Agreed, we have changed the last paragraph of the discussion so that it is more connected to what is discussed in the earlier paragraphs. We added discussion "about the coupling between N and P cycles and how this could be affected by climate change or disturbances and in turn affect ecosystem status and processes", as suggested.

p. 9, l. 6: "variation in the vegetation", be more precise (which parameter, sense of variation).

**Response:** Agreed and changed to "changes in plant species composition"

p. 9, l. 10 and 17: "vegetation dynamics", this is not what you study here, change to "vegetation community composition" or something like that.

**Response:** Agreed and changed.

p. 9, l. 12: change "has been discovered" to "was found".

**Response:** Agreed and changed.

Fig. 2: draw or remind in the title where tree cover was described.

**Response:** We have now added to the caption that trees were measured from the whole 30 x 30 m area that the clusters delineate. We also drew marks to the figure to make it clearer where the outer borders of the 30 x 30 m area were.

Fig. 3: would it be possible to also represent tree cover species (or different groups based on dominant species)? or age? or geology? Is this graph representing between- plot variability (i.e. you first calculated the mean for each plot and made the boxplots with those means) or a mix between within- and between-plot variability (i.e. you took all sub-plots values to make the boxplot)? It would be interesting to compare between- and within-plot variabilities.

Fig. 4: same comments as for Fig. 3.

**Response:** We have replaced these figures with box plots, which show the variation of soil nutrients between and within plots grouped based on their dominant tree species.

Fig. 5: What is the correlation coefficient calculated? (Pearson? Other?) Precise what is the "bottom layer". Why not to call this layer "moss lichen"? It would be clearer.

**Response:** We added that Pearson correlations were used. The bottom layer includes mosses and lichen, so we re-named this layer "moss lichen" and redid the figure.

Fig. 6: what is the criteria to define "the most abundant species"?

**Response:** We added here that most abundant species here means those species, which have the highest % coverage.

Fig. 6, l. 8: replace "generic" by "genera". Start the last sentence of the figure legend by "In (a),"

**Response:** Agreed and done.

Fig. 7: remind which soil layer was considered for this analysis (I am assuming it's O).

**Response:** Yes, it is O layer and we added this information to the figure caption.

Table 1: "degree days", shouldn't that be called "sum of degree days"? What is the unit?

**Response:** Degree days here mean the growing degree days, whose unit is GDD or °C days. We changed it to "growing degree day sum" as well added how it is calculated:

*"Growing degree day sum was calculated as the average daily temperature (average of daily maximum and minimum temperatures) above 5 °C base temperature, accumulated on a daily basis over the year. Negative values are treated as zeros and ignored."*

Table 2: the classical ordering would be C, N, P, K, C:N, N:P. For numbers (mean and sd), provide the same number of digits after the dot for each column (one is enough) and align the numbers to the right to ease comparison of lines. Why not to put K in the same unit as the others?

**Response:** Agreed, we have made these changes.

Table 3: did you estimate the whole aerial volume of trees or just the trunk volume? Make three sub-columns for each species abundance. Add in this table tree height, diameter,...

**Response:** We estimated the trunk volume. We made the suggested changes.

Table 4: remind the first seven lines are soil values.

**Response:** Agreed and done.

Tables A2–A4: From which statistical test are these table issued? These tables hardly help to address your objectives.

**Response:** We have deleted these tables and replaced them with a table which includes the fixed effects (from the linear mixed effect models) and their Chisq values, p-values and pseudoR2 values.

Table B1: title: change to "Statistically significant differences between needle age group by species". From which test?

**Response:** We changed the title and added that these are from the one-way ANOVA, with Tukey's HSD for post hoc. We will add this information to the caption.

Appendix C: precise "(% of surface area)".

**Response:** Agreed and done.

[revised manuscript text omitted]

.

---

## Author Response (AR2)

Dear Yakov Kuzyakov,

Thank you for your comments on the previous version of our manuscript. We appreciate the suggestions and have done the changes. Our comments are marked here in blue.

- remove "We studied, we measured, we established, we ..." from the Abstract

This has been done.

- finish the Abstract with a generalizing conclusion sentence

We added the following sentence to the end of the abstract: "The possible mining effects together with climate change can have an influence on the release of nutrients to plants, which may lead to alterations in the vegetation community composition in the study region."

- do not repeat the same word in one sentence: e.g. 1st sentence: soil soil soil

We split the first sentence of the abstract to two sentences and changed the wording as follows: "The relationship of the community composition of forest vegetation and soil nutrients were studied near Sokli phosphate ore in northern Finland. Simultaneously the effects of the dominant species and the age of trees, rock parent material and soil layer on these nutrients were examined.

- the right part of the Fig 1 is useless - all people know where Finland is located

We removed the right part of the figure.

- on Figs 3 and 4 you have strange data anrrangeement: Usually dependent variable is on the Y axis

The figures have been changed so that the dependent variable is on the y-axis. We changed the plot-wise figures in Appendix C (Figs. C2 and C3) to the similar form.

- the text on Figs 6 & 7 is too small - not possible to read

We replaced these figures with ones that had a bigger font size.

- legend to Table 4 is shifted - check similar problems in the whole text

This has been checked.

- Tables 1, 4 and 5 can be moved to Supplementary Materials / Appendix

This has been done.

Kind regards, Laura Matkala

[revised manuscript text omitted]